# Phytochemical Profiling, Antioxidant and Cognitive-Enhancing Effect of *Helichrysum italicum* ssp. *italicum* (Roth) G. Don (Asteraceae)

**DOI:** 10.3390/plants12152755

**Published:** 2023-07-25

**Authors:** Reneta Gevrenova, Ivanka Kostadinova, Alexandra Stefanova, Vessela Balabanova, Gokhan Zengin, Dimitrina Zheleva-Dimitrova, Georgi Momekov

**Affiliations:** 1Department of Pharmacognosy, Faculty of Pharmacy, Medical University, 1000 Sofia, Bulgaria; rgevrenova@pharmfac.mu-sofia.bg (R.G.); vbalabanova@pharmfac.mu-sofia.bg (V.B.); 2Department of Pharmacology, Pharmacotherapy, and Toxicology, Faculty of Pharmacy, Medical University, 1000 Sofia, Bulgariaastefanova22@gmail.com (A.S.); gmomekov@pharmfac.mu-sofia.bg (G.M.); 3Physiology and Biochemistry Research Laboratory, Department of Biology, Science Faculty, Selcuk University, Konya 42130, Turkey; gokhanzengin@selcuk.edu.tr

**Keywords:** *Helichyrsum*, antioxidant, memory function, phenolics, enzyme inhibition

## Abstract

This study aimed at the evaluation of the antioxidant and cognitive-enhancing effect of methanol–aqueous extract from *Helichrysum italicum* ssp. *italicum* aerial parts. Significant radical scavenging activity (110.33 ± 3.47 and 234.70 ± 5.21 mg TE/g for DPPH and ABTS) and reducing power (354.23 ± 17.51 and 210.24 ± 8.68 mg TE/g for CUPRAC and FRAP) were observed. The extract showed average acetylcholinesterase and low butyrylcholinesterase inhibitory potential. *H. italicum* extract (200 mg/kg/po) administered in combination with galantamine (3 mg/kg/po) for 12 days significantly improved the memory and learning process compared with galantamine alone in the passive avoidance test. The effect was comparable to that of *Ginkgo biloba* extract (100 mg/kg/po). In deep secondary metabolite annotation of the extract by UHPLC-HRMS, more than 90 hydroxybenzoic and hydroxicinnamic acid-glycosides, phenylethanoid glycosides, a series of acylquinic and caffeoylhexaric acids, methoxylated derivatives of scutellarein, quercetagetin and 6-hydroxyluteolin, and prenylated phloroglucinol-α-pyrones were reported for the first time in *H. italicum*. Fragmentation patterns of four subclasses of heterodimer-pyrones were proposed. In-depth profiling of the pyrones revealed 23 compounds undescribed in the literature. Pyrones and acylphloroglucinols together with acylquinic acids could account for memory improvement. The presented research advanced our knowledge of *H. italicum*, highlighting the species as a rich source of secondary metabolites with cognitive-enhancing potential.

## 1. Introduction

*Helichrysum italicum* (Roth) G. Don (curry plant, immortelle) is an aromatic perennial subshrub belonging to the family Asteraceae, subfamily Asteroideae, and tribe Gnaphalieae. The genus *Helichrysum* Mill. comprises approximately 600 species distributed in central Asia, India, Africa, including Madagascar, and the Mediterranean Basin [1]. The genus has a high occurrence in the Mediterranean areas of Europe. Based on the morphological features, genetic variation, and geographical distribution, *Helichrysum italicum* is divided into six subspecies (ssp.): *italicum*, *microphyllum*, *picardii*, *siculum*, *serotinum,* and *tyrrhenicum* [2]. The name of the genus is derived from the Greek words “helios” and “chryos”, which mean “sun” and “gold”, respectively. This nomenclature is due to the inflorescences of a bright yellow color typical for the taxon [3]. The common name “immortelle” is related to the everlasting flowers that retain their form and color when dried and thus are used in dry bouquets [4].

*H. italicum* essential oil is wealthy in various terpenes and mono- and sesquiterpenes with high contribution to the biological activity [2,5]. Recently, the chemical composition of the essential oil of flowering aerial parts of *H. italicum* subsp. *italicum* cultivated in central Italy was analyzed by gas chromatography–mass spectrometry. Seventy-eight components were identified and quantified where neryl acetate revealed the largest relative abundance in the composition (15.75% of the oil), followed by *α*-pinene (8.21%), 4,6,9-trimethyl-8-decene-3,5-dione (italidione I) (7.34%), curcumene and *β*-selinene (5.37%), *γ*-curcumene (4.83%), nerol (4.75%), *α*-selinene (4.68%), limonene (4.55%), linalool (4.42%), and 2,4,6,9-tetramethyl-8-decene-3,5-dione (italidione II) (4.26%). As a result, the oil inhibited in vitro collagenase and elastase activities, with IC50 values of 36.99 ± 1.52 and 135.43 ± 6.32 μg/mL, respectively [6]. Neryl acetate is also the predominant compound, with amounts from 15.8% (from plants in the stage of early shoots) to 42.5% (in the full flowering period) in the oil from the *H. italicum* subsp. *italicum* growing in Corsica [7]. However, further studies showed that phenolic and oxygenated compounds contributed to the major components. The reported secondary metabolites from the genus can be categorized into six structural types: flavonoids and chalcones, phenolic acids, terpenes and essential oils, pyrones (both homo- and heterodimeric), benzofurans (bitalin esters), and phloroglucinols [8] consisting mainly of two types of substituents, a prenyl/geranyl group and an acyl group. The most common acyl substituents are methyl, isopropyl, and 2-methylbutanoyl. Various phenolic metabolites were previously isolated and identified including flavonoids, acetophenones, phloroglucinols, tremetones, coumarins and coumarates, phenolic acids, and esters, with their derivatives [5]. More than 100 different compounds were formerly reported as part of the *H. italicum* species with prominent biological activities including the following phenolic acids: hydroxycinnamic acids and derivates, caffeic acid and derivates, and isomers of monocaffeoylquinic acids (CQAs), di- or tri-CQAs, and their glucosides and esters; hydroxybenzoic acids, flavonoids, and their glucosides; and pyrone derivates [9]. Some of the most promising and notable bioactive compounds of *H. italicum* are caffeic acid, chlorogenic acid, pinocembrin, quercetin, naringenin, gnaphaliin, luteolin, tiliroside, arzanol, and ursolic acid [10]. High-resolution mass spectrometry (HRMS) coupled with ultra-high-performance liquid chromatography (UHPLC) and data dependent MS/MS analyses provide very valuable information on secondary metabolites for in-depth metabolite annotation studies [11].

Oxidative stress is an imbalance between reactive oxygen species (ROS) generation and antioxidant defense systems, which are implicated in different pathways of injury in the development of various disorders, including neurodegenerative disorders and aging [12]. As antioxidants are substances that are efficient to scavenge ROS and decrease oxidative damage, these compounds have been studied for therapeutic approaches to many different diseases. Moreover, a deficiency of antioxidants such as vitamins C and E has been associated with cognitive disorders [13]. Therefore, many natural products, especially polyphenols with antioxidant properties could be useful in cognitive decline or neurodegenerative diseases such as Alzheimer’s disease [14]. Recently, in vitro DPPH and ABTS antioxidant activity of both essential oils and methanolic extracts of herbs and inflorescences of *H. italicum* has been reported [15]. Results revealed that methanolic extract and essential oil obtained from the aerial parts indicated a higher potential than those originating from the inflorescences. The stronger antioxidant potential may be related to the content of phenolics, especially rutin and chlorogenic acid, known as strong antioxidants [15].

*H. italicum* is commonly used in the traditional medicine with influences on allergies, colds, coughs, skin, inflammation, infections, and sleeplessness [5]. Previous ethnopharmacological surveys showed the use of *H. italicum* related to different respiratory, digestive, and inflammatory conditions. Its effect on skin conditions such as hematoma, scars, and even psoriasis are also investigated [2]. Several surveys documented the potential use and influence of *H. italicum* on obesity, metabolic syndrome, and type 2 diabetes mellitus. Additionally, clinical data revealed that *H. italicum* subsp. *italicum* infusion acutely increased the fat oxidation and resting energy expenditure and induced the metabolic effects [16]. A clinical trial investigated the result of *H. italicum* essential oil on mental exhaustion and moderated burnout and revealed a reduction in the perceived level of mental fatigue/burnout [17].

Despite the extensive investigation of the chemical composition and biological activities, there are limited data on the ability of *H. italicum* extracts to improve memory and learning. Hence, the aim of this study was to analyze the secondary metabolites in *H. italicum* var. *italicum*, in the light of its antioxidant and cognitive-enhancing effect. In addition, we hypothesized that the improvement of the learning and memory processes of *H. italicum* methanol–water extract could be attributed to its powerful antioxidant and moderate acetylcholinesterase inhibitory activity.

## 2. Results and Discussion

A flow chart for the design of the experiment was presented in Figure 1. *H. italicum* aerial parts were extracted with 80% methanol and lyophilized. The dissolved lyophilized extract was first injected in UHPLC-HRMS. Data were acquired using the data-dependent acquisition mode and then converted through a MZmine 2.53 software processing. After obtaining the UHPLC-HRMS profiling, in negative and positive ion modes, extracted chromatograms and their MS/MS spectra followed. Based on the literature data, using the taxonomy as a filter and comparing it with authentic standards, dereplication/annotation of the peaks was carried out. In parallel, spectrophotometric assays were conducted to determine antioxidant and cholinesterase inhibitory potential. Finally, the cognitive-enhancing effect was determined using an in vivo passive avoidance test.

### 2.1. UHPLC-HRMS Profiling of H. italicum Extract

Herein, a comprehensive UHPLC-HRMS analysis of *H. italicum* methanol–aqueous extract was performed yielding the identification/annotation of more than 160 secondary metabolites (Table 1 and Appendix A). The total ion chromatogram (TIC) in the negative ion mode of the studied extract was depicted in Figure 2.

For the first time, 12 phenolic acids and derivatives and coumarins, 14 acylquinic acids, 15 acylhexaricic acids, 22 flavonoid aglycones and glycosides, and 28 heterodimer-pyrones were reported in *H. italicum* (Table 1, Appendix A). To the best of our knowledge, phenylethanoid glycosides, a series of hydroxybenzoic and hydroxicinnamic acid–glycosides, hydroxycinnamoyl hexoses (sugar esters), caffeoyl-hydroxydihydrocaffeoylquinic, malonyl-dicaffeoylquinic, *p*-coumaroyl-caffeoylquinic and tricaffeoylquinic acids, and dicaffeoylquinic acid-hexoside were reported for the first time. Within a group of acylhexaric acids, hydroxybutanyl-tricaffeoylhexaric, isobutanyl-tricaffeoylhexaric, and 2-methyllbutanyl/isovaleryl-tricaffeoylhexaric acids were not previously reported in the literature. A series of methoxylated derivatives of scutellarein, quercetagetin, and 6-hydroxyluteolin together with flavonol-hydroxycinnamoylglycosides represent new secondary metabolites in the species. This study is the first attempt at an in-depth characterization of phloroglucinol α-pyrones by hyphenated technique LC-MS allowing for the annotation of a series of heterodimers not previously reported even in the genus *Helichrysum*. It is worth noting that the fragmentation patterns of four subclasses of pyrones are suggested (Appendix A, Figure 1 and Appendix A).

### 2.2. Hydroxybenzoic, Hydroxycinnamic Acids and Their Derivatives

A variety of hydroxybenzoic and hydroxycinnamic acid hexosides was tentatively identified including **1**–**3**, **6**–**10**, **16**–**18**, **20**, **21**, and **23** (Table 1 and Appendix A).

MS/MS spectra of sugar esters hydroxybensoyl-hexose (**5**), vanillyl-hexose (**11**), and syringoyl-hexose (**13**) were acquired. In contrast to the analogous hexosides, the precursor ions afforded fragment ions resulting from the hexose cross ring cleavages as ^0.4^Hex (−60 Da), ^0.3^Hex (−90 Da), and ^0.2^Hex (−120 Da) [18] (Figure 3).

Based on the comparison with the retention times and fragmentation patterns of reference standards, six hydroxybenzoic acids (**4**, **14**, **15**, **29**, **30**, and **34**) and four hydroxycinnamic acids (**22**, **26**, **28**, and **31**) together with quinic acid (**19**) and *p*-hydroxyphenylacetic (**25**) were identified in the extract (Appendix A).

Within this group, **26** (caffeic acid) was the main compound in the studied extract, together with **19**, **24**, and **25** (Appendix A). Although hydroxybenzoic and hydroxycinnamic acids were present in their free form, herein, a large number of phenolic acids hexosides were revealed in *H. italicum* for the first time. The extracted ion chromatograms of hydroxybenzoic and hydroxycinnamic acids and derivatives showed that the *H. italicum* profile was dominated by caffeic acid (**26**) (13.95%), protocatechuic acid-*O*-hexoside (**2**) (13.10%), dehydrochorismic acid (**27**) (7.88%), and vanillic acid (**30**) (5.20%) together with 4-hydroxybenzoic acid (**14**) (4.74%) and protocatechuic acid (**4**) (4.44%) (Appendix A).

### 2.3. Acylquinic Acids (AQA)

Ten mono-, nineteen di-, and one triacylquinic acids (AQAs) were identified or annotated in the *H. italicum* extract (Table 1 and Appendix A). Fragmentation patterns were consistent with those reported elsewhere [19,20,21]. Thus, **38**/**41**, **42**/**48**, and **46** were assigned to 5-caffeoyl-, 5-*p*-coumaroyl, and 5-feruloylquinic acid, respectively, as the base peak was observed at *m/z* 191.055, while 3-caffeoyl-(**36**) and 3-feruloylquinic acid (**40**) were evidenced by the abundant ions at *m/z* 179.034 (65%) and 193.050 (100%), respectively (Appendix A).

diAQA consisted of the following subclasses: dicaffeoylquinic acids (*di*CQA) (**44**, **51**, **52**, **54**, and **56**), dicaffeoylquinic acid malonyl (**53**, **55**, **57**, and **58**), feruloyl-caffeoylquinic acids (FCQA) (**60**, **61**, **63**, and **64**), *p*-coumaroyl-caffeoylquinic acids (*p*-CoCQA) (**59** and **62**), and hydroxydihydrocaffeoyl-caffeolylquinic acids (HC-CQA) (**43**, **45**, and **47**) (Appendix A).

Compounds **59** and **60** afforded prominent ions at *m/z* 337.093 (81%) and 367.103 (100%), respectively, indicating a loss of caffeoyl residue before the *p*-coumaroyl (**59**) and feruloyl (**60**) moiety. Furthermore, both compounds yielded fragment ions at *m/z* 163.039 (100%) and 193.050 (93%), respectively, as was observed in 3-AQA together with *m/z* 119.049 [*p*-coumaric acid-H-CO_2_]^−^ (36%) (**59**) and 134.036 [ferulic acid-H-CH_3_-CO_2_]^−^ (63%) (**60**) (Appendix A, Figure 3). Thus, **59** and **60** were ascribed as 3-*p*-Co-5CQA and 3F-5CQA, respectively.

Based on the distinctive “dehydrated” peak at *m/z* 173.045 [quinic acid-H-H_2_O]^−^, **62** and **63** were annotated as vicinal 4-*p*-Co-5CQA and 4F-5CQA. Likewise, the vicinal *di*CQA 3, 4-diCQA (**51**), and 4,5-diCQA (**56**) were also defined (Appendix A). The assignment of *di*CQA malonyl esters (**53**, **55**, **57**, and **58**) was suggested by the transitions resulting from the loss of 86.001 Da (C_3_H_2_O_3_) or the malonyl group. Thus, **53** produced 601.121→515.120 together with the base peak at *m/z* 233.066, indicating concomitant loss of two caffeoyl moieties and CO_2_ (Appendix A, Figure 3). Peak **50** afforded a precursor ion at *m/z* 677.173 (calc. for C_31_H_33_O_17_) together with the transitions at *m*/*z* 677.173→515.141→353.088→191.055 resulting from the losses of two caffeoyl residues and hexose unit, respectively (Appendix A). The 1,3-*di*substituted quinic acid skeleton was discernible by the ions at *m*/*z* 191.055 (82%), 179. 034 (99%), and 135.044 (100%). Thus, **50** was ascribed as 1,3-*di*caffeoylquinic acid-hexoside.

The most lipophilic AQA was **65**; 3,4,5-triCQA was discernable by the prominent ions at *m/z* 179.034, 173.045, and 135.044, as was observed in the 3,4-disubstituted quinic acid skeleton. Among acylquinic acids, the predominant compounds in *H. italicum* extract were 3,4-dicaffeoylquinic acid (**51**) (18.99%) and 3,5-dicaffeoylquinic acid (**52**) (18.92%), followed by 1,5-dicaffeoylquinic acid (**54**) (15.51%) and 4,5-dicaffeoylquinic acid (**56**) (8.80%) (Appendix A).

### 2.4. Caffeoylhexaric Acids

Key points in the acylhexaric acids annotation were the subsequent losses of one (**66**–**69**), two (**70**–**74**), and three (**75**–**78**) caffeoyl residues (Table 1 and Appendix A). Thus, the base peak in the MS/MS spectra was consistent with [hexaric acid (HA)-H]^−^ at *m*/*z* 209.030 (C_6_H_9_O_8_) supported by the series of prominent ions at *m*/*z* 191.019 [HA-H-H_2_O]^−^, 147.029 [HA-H-H_2_O-CO_2_]^−^, 129.018 [HA-H-2H_2_O-CO_2_]^−^, 111.007 [HA-H-3H_2_O-CO_2_]^−^, and 85.028 [HA-H-2H_2_O-2CO_2_]^−^ (Table S1) [22]. Compounds **79** and **80** shared the same [M-H]^−^ at *m*/*z* 781.162 (calc. for C_37_H_34_O_19_ (Appendix A, Figure 3). They produced the indicative fragment ions at *m*/*z* 353.052 [M-H-2caffeoyl-C_4_H_8_O_3_]^−^ and 191.019 [M-H-3caffeoyl-C_4_H_8_O_3_]^−^ resulting from the concomitant losses of caffeoyl residues and hydroxybutyric acid. Caffeoyl moiety was suggested by the fragment ions at *m*/*z* 179.0.4 [caffeic acid (CA)-H]^−^, 161.023 [(CA-H)-H_2_O]^−^, and 135.044 [(CA-H)-CO_2_]^−^. Compounds **79** and **80** were ascribed as isomeric hydroxybutanyl-tricaffeoylhexaric acids. Similarly, **81** ([M-H]^−^ at *m*/*z* 765.167, C_37_H_34_O_18_) was assigned to isobutanyl-tricaffeoylhexaric acids as indicated by the transition 279.072→191.019, suggesting a loss of 88.054 Da (calc. for C_4_H_8_O_2_) or isobutyric acid. In the same manner, 2-methylbutanyl/isovaleryl residue in **82** was deduced from the prominent ions at 293.088 [M-H-3caffeoyl]^−^ and a subsequent loss of 102.069 Da (calc. for C_5_H_10_O_2_) or 2-methylbutiric acid/isovaleric acid at *m*/*z* 191.019 (Appendix A). Thus, **82** was ascribed as 2-methylbutanyl/isovaleryl-tricaffeoylhexaric acid.

Among caffeoylhexaric acids, tricaffeoylhexaric acid 1 (**75**) (22.01%), together with its isomers **76** (11.93%), **77** (11.11%), and **78** (10.92%), appeared to be dominant for *H. italicum* extract (Appendix A).

### 2.5. Flavonoids

Typical ions of the *O*-glycosyl flavonoid pathway were produced in a variety of flavones and flavonol-hexosides including **83**, **85**–**87**, **89**–**91**, **94**, **96**, and **97** (Table 1 and Appendix A).

In (-) ESI mode, the precursor ions provided the relevant ions at *m/z* 317.030 (myricetin), 301.035/300.028 (quercetin), 285.040 (luteolin and kaempferol), 331.046 (patuletin), 329.066 (jaceosidin), 315.047 (nepetin and isorhamnetin), and 299.056 (hispidulin). A 6-methoxylated flavonoid skeleton was discernable by the RDA ions at *m/z* 165.990 (^1,3^A^—^CH_3_), 164.982 (^1,3^A^—^CH_4_), 163.002 (^1,3^A^—^H_2_O), and 136.987 (^1,3^A^−^-CH_4_-CO) [13,14]. In addition, a low abundant ion at 181.013 (^1,3^A^−^) was registered in the fragmentation pattern of patuletin and 6-methoxykaempferol.

The two compounds **88** and **92** shared the same [M-H]^−^ at *m/z* 505.099, yielding prominent fragment ions at *m/z* 463.086.094 [M-H-C_2_H_2_O]^−^ (**88**) and *m/z* 301.035 [M-H-(C_2_H_2_O+Hex)]^−^, indicating a subsequent loss of an acetyl group (42 Da) and hexosyl (162 Da) (Appendix A). In addition to the aglycone quercetin (Y_0_^−^), an abundant radical aglycone [Y_0_-H]^−•^ was also formed, suggesting a 3-*O*-glycosidic bond [23]. In the same manner, kaempferol 3-*O*- and myricetin 3-*O*-acetylhexoside were depicted in flavonoid profiling.

The two compounds **109** ([M-H]^−^ at *m/z* 609.125) and **111** ([M-H]^−^ at *m/z* 639.136) shared the same fragmentation patterns yielding prominent fragment ions at *m/z* 447.092 (**109**) and 477.104 (**111**), respectively, resulting from loss of a caffeoyl moiety (162 Da, C_9_H_6_O_3_) (Appendix A). Moreover, caffeoyl moiety was evidenced by the fragment ions at *m/z* 179.034 [(caffeic acid-H)]^−^ (**109**), 161.023 [(caffeic acid-H)-H_2_O]^−^, and 135.044 [(caffeic acid-H)-CO_2_]^−^. Consequently, **109** was ascribed as kempferol *O*-caffeoylhexoside, while **111** was annotated as isorhamnetin *O*-caffeoylhexoside.

In the same way, **98/100**, **103/108**, and **107/110** were assigned to coumaroyl esters of quercetin-, kaempferol, and isorhamnetin *O*-hexoside (Appendix A). The commonly found losses of 146 Da (C_9_H_6_O_2_) and 308 Da (C_15_H_16_O_7_) in the fragmentation patterns of the aforementioned compounds accompanied with the ions at *m/z* 163.039 [coumaric acid-H]^−^ and 145.028 [(coumaric acid-H)-H_2_O]^−^ confirmed the presence of a coumaroyl moiety. One feruloyl ester of quercetin-hexoside (**99**) was evidenced on the base of the loss of 176 Da (C_10_H_8_O_3_) at *m/z* 463.088 023 and 338 Da (C_16_H_18_O_8_) at *m/z* 301.0349 (Appendix A, Figure 3). A malonyl ester **93** was discernable by the transition [M-H]^−^→Y_0_ resulting from the indicative loss of 248.046 Da (C_9_H_12_O_8_).

The approach for 6-methoxylated flavonoids annotation was delineated elsewhere [12,13,14]. Among them, five methoxylated derivatives of quercetagetin (**112**, **119**, and **122**), scutelarein (**113**), and 6-hydroxyluteolin (**117**) were described (Appendix A).

As an example, compound **119** ([M-H]^−^ at *m/z* 359.077, C_18_H_15_O_8_) was used to illustrate the fragmentation pattern of 6-methoxylated quercetagetin derivatives (Appendix A). In (-) ESI-MS/MS, **119** afforded typical radical losses at *m/z* 344.054 [M-H-•CH_3_]^−^, 329.030 [M-H-2•CH_3_]^−^, and 314.007 [M-H-3•CH_3_]^−^, accompanied with subsequent neutral losses at *m/z* 301.035 [M-H-2•CH_3_-CO]^−^, 286.012 [M-H-3•CH_3_-CO]^−^, 258.017 [M-H-3•CH_3_-2CO]^−^, 230.021 [M-H-3•CH_3_-3CO]^−^, and 202.026 [M-H-3•CH_3_-4CO]^−^ (Figure 3). The precursor ion yielded a series of low abundant fragment RDA ions at *m/z* 165.990 (^1,3^A^−^-•CH_3_), 136.987 (^1,3^A^−^-CO-CH_4_), and 109.9997 (^1,3^A^−^-CO-CHO-CH_3_), indicating methoxylation in the A-ring (Ren et al., 2018). On the other hand, prominent fragment ions at *m/z* 163.038 (^1,3^B^−^-CH_2_) and ^1,3^B^−^ at *m/z* 148.016 (^1,3^B^−^-•CH_3_-CH_2_) (Appendix A) pointed out a dimethoxylated RDA ion ^1,3^B. Thus, **119** and **122** were ascribed as quercetagetin-3,6,3’(4’)-trimethyl ether. In line with Kramberger et al. [7], either initial RDA ions or their derivatives were not registered in **118** (gnaphaliin) with *m/z* [M-H]^−^ at *m/z* 313.072, C_17_H_13_O_6_ (Appendix A). In addition to the abundant ions at *m/z* 298.048 and 283.025 resulting from the radical losses, a series of neutral losses (CO and CO_2_) was also generated at *m/z* 199.039 [M-H-2•CH_3_-3CO]^−^, 183.044 [M-H-2•CH_3_-2CO-CO_2_]^−^, and 139.054 [M-H-2•CH_3_-2CO-2CO_2_]^−^.

In the MS/MS spectra of pinocembrin (**123**) and galangin (**124**), the corresponding ^1,3^B ions were not found—both compounds have no substitution on the B-ring.

Overall, galangin methyl ether (**126**) (18.43%) and gnaphaliin 2 (**125**) (14.71%) were found to be the predominant flavonoids followed by hyperoside (**85**) (7.44%), quercetin 7,3’(4’)-dimethyl ether (**120**) (4.80%), quercetin *O*-coumaroylhexoside isomer (**98**) (4.68%), and quercetin *O*-coumaroylhexoside 2 (**100**) (4.67) (Appendix A).

### 2.6. Prenylated Phloroglucinol α-Pyrones

Pyrones refer to the special class heterodimers consisting of prenylated phloroglucinyl and α-pyrone [24]. In the MS/MS spectra, all the [M-H]^−^ lose α-pyrone moieties and form abundant peaks [M-H-pyrone]^−^ (Table 1 and Appendix A). Hence, the characteristic neutral losses of *m/z* 140.048 (C_7_H_8_O_3_), 154.063 (C_8_H_10_O_3_), 168.079 (C_9_H_12_O_3_), 182.095 (C_10_H_14_O_3_) Da was employed to recover α-pyrone residue in each derivative. Pyrones typically cleave at C-7 and accordingly lose additional carbon (−12 Da) as follows: *m/z* 152.048 (C_8_H_8_O_3_), 166.063 (C_9_H_10_O_3_), 180.079 (C_10_H_12_O_3_), and 194.095 Da (C_11_H_14_O_3_) (Appendix A). Diagnostic fragment ions for the corresponding pyrone units were registered in the (-) ESI-MS/MS spectra at *m/z* 139.039 (C_7_H_7_O_3_), 153.054 (C_8_H_9_O_3_), 167.070 (C_9_H_11_O_3_), and 181.086 (C_10_H_13_O_3_). In addition, all pyrone units provided [Pyrone-H-CO_2_]^−^ at *m/z* 95.048, 109.064, 123.080, and 137.096, respectively. Accordingly, in the (+) ESI-MS/MS spectra, fragment ions at *m/z* 141.055 (C_7_H_9_O_3_), 155.070 (C_8_H_11_O_3_), 169.086 (C_9_H_13_O_3_), and 183.102 (C_10_H_15_O_3_) were produced.

Thus, a methyl group at C-11 in the α-pyrone unit was deduced in arenol (**131**) and **148**. The majority of the pyrones were assigned to phloroglucinyl α-pyrones possessing an ethyl group at C-11. An isopropyl group was evidenced at C-11 of **130**, **139**, **144**, and **157**, while 1-methyl-propyl moiety was found in **143**, **149**, and **150**. Accordingly, four subclasses of pyrones were suggested as follows: methylpyrones (MP), ethylpyrones (EP), isopropylpyrones (IPP), and 1-methyl-propylpyrones (MPP). The tentative structures of phloroglucinyl α-pyrones are depicted in Appendix A.

Exemplified by **135** ([M-H]^−^ at 401.160), in (-) ESI-MS, phloroglucinol skeleton gave prominent fragment ions at *m/z* 205.086 [M-H-C_8_H_10_O_3_-C_2_H_2_O]^−^, 191.107 [M-H-C_8_H_10_O_3_-2CO]^−^, and 166.026 [M-H-C_9_H_10_O_3_-C_5_H_9_]^−^, indicating a loss of an acyl group at C-1 and a prenyl chain at C-3, respectively (Figure 4 and Appendix A). On the other hand, in (+) ESI-MS/MS, the base peak at *m/z* 181.050 resulted from the typical concomitant losses of (α-pyrone-C, C_9_H_10_O_3_) and C_4_H_8_. Thus, **135** was assigned as arzanol [25,26]. An MS/MS spectrum of **141** was acquired. [M-H]^−^ at *m/z* 415.175 (calc. for C_23_H_27_O_7_) indicated 14 Da higher than that of **135**, suggesting an additional methylene group and an oxopropyl moiety at C-1. This assumption was confirmed by the indicative fragment ions in (+) ESI-MS at *m/z* 417.190 [M+H-C_4_H_8_]^+^ and 195.065 [M+H-C_9_H_10_O_3_-C_4_H_8_]^+^. In the same manner, in (+) ESI-MS, **146** gave [M+H]^+^ at *m/z* 431.206 (calc. for C_24_H_31_O_7_) and prominent fragment ions at *m/z* 375.143 [M+H-C_4_H_8_]^+^ and 209.081 [M+H-C_9_H_10_O_3_-C_4_H_8_]^+^ consistent with a prenyl group at C-3 and 2-methyl-oxopropyl residue at C-1 as was seen in 6-*O*-desmethylauricepyrone isolated from *H. oocephalum* and *H. stenopterum* DC (Appendix A) [24]. Compound **151** yielded [M-H]^−^ at *m/z* 443.208 (C_25_H_31_O_7_), indicating an additional methylene group compared to **146**. Ethylpyrone moiety was evidenced by the typical neutral losses at *m/z* 291.159 [M+H-C_8_H_10_O_3_]^+^ and *m/z* 279.159 [M+H-C_9_H_10_O_3_]^+^ accompanied by the subsequent loss of 56 Da (C_4_H_8_) at *m/z* 235.096 and 223.0963, respectively (Appendix A). Thus, 2-methyl-oxobutyl residue was dedicated at C-1, and **151** could be ascribed as 23-methyl-6-*O*-desmethylauricepyrone isolated previously from *H. stenopterum* and *H. oocephalum* [24,27]. The abovementioned functional groups at C-1 of phloroglucinol moiety were consistent with those evidenced in the heterodimeric pyrones from *H. italicum* subsp. *microphyllum* [6] and the prenylated phloroglucinols isolated from *H. paronichioides* [28].

**Table 1 plants-12-02755-t001:** Secondary metabolites in *Helichrysum italicum* methanol–aqueous extract.

№	Identified/Tentatively Annotated Compound	Molecular Formula	Exact Mass [M-H]^−^	t_R_ (min)	Δ ppm	Level of Confidence [21,29]	References
**Hydroxybenzoic, hydroxycinnamic, and phenylethanoid glycosides**							
1.	hydroxybenzoic acid-*O*-hexoside	C_13_H_16_O_8_	299.0778	1.28	−3.279	2	[9]
2.	protocatechuic acid-*O*-hexoside	C_13_H_16_O_9_	315.0727	1.68	0.115	2	[9]
3.	vanillic acid-*O*-hexoside	C_14_H_18_O_9_	329.0875	1.75	−0.198	2	[9]
4.	protocatechuic acid ^a^	C_7_H_6_O_4_	153.0181	2.02	−1.392	1	[26]
5.	hydroxybenzoyl hexose ^b^	C_13_H_16_O_8_	299.0778	2.04	0.002	2	
6.	protocatechuic acid-*O*-hexoside isomer	C_13_H_16_O_9_	315.0727	2.11	1.729	2	
7.	*p*-hydroxyphenylacetic acid *O*-hexoside ^b^	C_14_H_18_O_8_	313.0932	2.12	0.988	2	
8.	syringic acid-*O*-hexoside ^b^	C_15_H_20_O_10_	359.0985	2.24	0.036	2	
9.	caffeic acid-*O*-hexoside	C_15_H_18_O_9_	341.0871	2.41	−1.980	2	
10.	hydroxybenzoic acid-*O*-hexoside isomer	C_13_H_16_O_8_	299.0778	2.44	0.800	2	
11.	vanillyl *O*-hexose ^b^	C_14_H_18_O_9_	329.0875	2.47	0.075	2	
12.	esculetin-*O*-hexoside ^b^	C_15_H_15_O_9_	339.0724	2.70	0.515	2	
13.	syringyl-*O*-hexose ^b^	C_15_H_20_O_10_	359.0984	2.75	−0.278	2	
14.	4-hydroxybenzoic acid ^a^	C_7_H_6_O_3_	137.0230	2.83	−10.928	2	[26] [9]
15.	gentisic acid ^a^	C_7_H_6_O_4_	153.0181	2.98	−8.051	1	
16.	*p*-hydroxyphenylacetic acid *O*-hexoside isomer ^b^	C_14_H_18_O_8_	313.0929	2.99	0.030	2	
17.	hydroxybenzoic acid-*O*-hexoside	C_13_H_16_O_8_	299.0778	2.99	−0.704	2	[9]
18.	caffeic acid *O*-hexoside	C_15_H_18_O_9_	341.0871	3.07	−0.602	2	
19.	quinic acid	C_7_H_12_O_6_	191.0549	3.18	−5.817	2	[26]
20.	caffeic acid-*O*-hexoside	C_15_H_18_O_9_	341.0871	3.27	−0.386	2	[9]
21.	coumaric acid-*O*-hexoside	C_15_H_18_O_8_	325.0930	3.34	−3.355	2	[26] [9]
22.	*p*-coumaric acid ^a^	C_9_H_8_O_3_	163.0389	3.35	−8.203	1	[26]
23.	vanillic acid *O*-hexoside isomer	C_14_H_18_O_9_	329.0875	3.37	0.252	2	[9]
24.	esculetin	C_9_H_6_O_4_	177.0193	3.46	4.982	2	[9]
25.	*p*-hydroxyphenylacetic acid ^a,b^	C_8_H_8_O_3_	151.0401	3.46	−9.252	2	
26.	caffeic acid ^a^	C_9_H_8_O_4_	179.0339	3.54	−6.044	1	[26]
27.	dehydrochorismic acid ^b^	C_10_H_8_O_6_	223.0248	3.73	−7.180	2	
28.	*m*-coumaric acid ^a,b^	C_9_H_8_O_3_	163.0389	4.56	−7.958	1	
29.	syringic acid ^a^	C_9_H_10_O_5_	197.0446	4.76	−4.703	1	[26]
30.	vanillic acid ^a,b^	C_8_H_8_O_4_	167.0338	4.78	−7.615	1	
31.	*o*-coumaric acid ^a,b^	C_9_H_8_O_3_	163.0389	5.01	−7.958	1	
32.	scopoletin	C_10_H_8_O_4_	191.0350	5.06	−5.088	2	[6]
33.	caffeic acid *O*-(hydroxyisovaleryl)-hexoside ^b^	C_20_H_26_O_11_	441.1402	5.58	0.125	2	
34.	salicylic acid ^a^	C_7_H_6_O_3_	137.0230	6.27	−10.344	1	26]
35.	caffeic acid *O*-(hydroxybensoyl)-hexoside ^b^	C_22_H_22_O_11_	461.1089	8.26	0.142	1	
**Mono-, di-, and triacylquinic acids**							
36.	neochlorogenic (3-caffeoylquinic) acid ^a,b^	C_16_H_18_O_9_	353.0867	2.35	0.495	1	
37.	3-*p*-coumaroylquinic acid	C_16_H_18_O_8_	337.0928	3.02	1.096	2	
38.	chlorogenic (5-caffeoylquinic) acid ^a^	C_16_H_18_O_9_	353.0874	3.18	0.665	1	[9,26]
39.	4-caffeoylquinic acid	C_16_H_18_O_9_	353.0878	3.35	−0.100	2	[9]
40.	3-feruloylquinic acid	C_17_H_20_O_9_	367.1034	3.43	-0.096	2	[26]
41.	5-caffeoylquinic acid isomer	C_16_H_18_O_9_	353.0874	3.88	0.580	2	[9]
42.	5-*p*-coumaroylquinic acid	C_16_H_18_O_8_	337.0928	3.96	0.829	2	[9]
43.	3-caffeoyl-5-hydroxy-dihydrocaffeoylquinic acid ^b^	C_25_H_26_O_13_	533.1288	4.03	0.236	2	
44.	1, 3- dicaffeoylquinic acid ^b^	C_25_H_24_O_12_	515.1190	4.14	0.137	2	
45.	3-caffeoyl-4- hydroxy-dihydrocaffeoylquinic acid ^b^	C_25_H_26_O_13_	533.1288	4.37	2.1	2	
46.	5-feruloylquinic acid	C_17_H_20_O_9_	367.1034	4.42	−0.096	2	[9]
47.	1-caffeoyl-3-hydroxy-dihydrocaffeoylquinic acid ^b^	C_25_H_26_O_13_	533.1288	4.45	12.710	2	
48.	5-*p*-coumaroylquinic acid isomer	C_16_H_18_O_8_	337.0928	4.63	0.028	2	[9]
49.	4-feruloylquinic acid ^b^	C_17_H_20_O_9_	367.1034	4.69	−0.260	2	
50.	1,3-dicaffeoylquinic acid-hexoside	C_31_H_34_O_17_	677.1723	5.15	0.956	2	[9]
51.	3,4-dicaffeoylquinic acid ^a^	C_25_H_24_O_12_	515.1190	5.71	0.254	1	[9,26]
52.	3,5-dicaffeoylquinic acid ^a^	C_25_H_24_O_12_	515.1189	5.85	0.370	1	[9]
53.	1,3-dicaffeoylquinic acid malonyl ^b^	C_28_H_26_O_15_	601.1199	5.95	1.292	2	
54.	1,5-dicaffeoylquinic acid ^a^	C_25_H_24_O_12_	515.1190	6.03	−0.096	1	[9]
55.	3, 5-dicaffeoylquinic acid malonyl ^b^	C_28_H_26_O_15_	601.1199	6.15	−0.039	2	
56.	4,5-dicaffeoylquinic acid	C_25_H_24_O_12_	515.1190	6.24	0.254	2	[9]
57.	3, 4-dicaffeoylquinic acid malonyl ^b^	C_28_H_26_O_15_	601.1199	6.32	0.577	2	
58.	4, 5-dicaffeoylquinic acid malonyl ^b^	C_28_H_26_O_15_	601.1199	6.51	0.677	2	
59.	3-*p*-coumaroyl-5-caffeoylquinic acid ^b^	C_25_H_24_O_11_	499.1251	6.53	2.535	2	
60.	3-feruloyl-5-caffeoylquinic acid	C_26_H_26_O_12_	529.1356	6.83	−0.849	2	[9]
61.	3-caffeoyl-5-feruloylquinic acid ^b^	C_26_H_26_O_12_	529.1356	6.90	0.776	2	
62.	4-*p*-coumaroyl-5-caffeoylquinic acid ^b^	C_25_H_24_O_11_	499.1252	6.94	−4.117	2	
63.	4-feruloyl-5-caffeoylquinic acid	C_26_H_26_O_12_	529.1356	7.09	−0.037	2	[9]
64.	4-caffeoyl-5-feruloylquinic acid ^b^	C_26_H_26_O_12_	529.1356	7.18	0.474	2	
65.	3,4,5-tricaffeoylquinic acid ^a^	C_34_H_30_O_15_	677.1528	7.78	1.605	1	
**Caffeoylhexaric acids**							
66.	caffeoylhexaric acid ^b^ 1	C_15_H_16_O_11_	371.0620	1.29	1.470	2	
67.	caffeoylhexaric acid ^b^ 2	C_15_H_16_O_11_	371.0620	1.69	−1.979	2	
68.	caffeoylhexaric acid ^b^ 3	C_15_H_16_O_11_	371.0620	1.99	−4.028	2	
69.	caffeoylhexaric acid ^b^ 4	C_15_H_16_O_11_	371.0620	2.41	−6.956	2	
70.	dicaffeoylhexaric acid 1	C_24_H_22_O_14_	533.0937	3.75	0.134	2	[9]
71.	dicaffeoylhexaric acid ^b^ 2	C_24_H_22_O_14_	533.0937	4.13	−8.213	2	
72.	dicaffeoylhexaric acid ^b^ 3	C_24_H_22_O_14_	533.0937	4.71	−14.066	2	
73.	dicaffeoylhexaric acid ^b^ 4	C_24_H_22_O_14_	533.0937	4.89	−14.516	2	
74.	dicaffeoylhexaric acid ^b^ 5	C_24_H_22_O_14_	533.0937	5.15	−8.457	2	
75.	tricaffeoylhexaric acid 1	C_33_H_28_O_17_	695.1254	5.89	0.788	2	[9]
76.	tricaffeoylhexaric acid ^b^ 2	C_33_H_28_O_17_	695.1254	6.32	1.219	2	
77.	tricaffeoylhexaric acid ^b^ 3	C_33_H_28_O_17_	695.1254	6.44	1.133	2	
78.	tricaffeoylhexaric acid ^b^ 4	C_33_H_28_O_17_	695.1254	6.53	2.011	2	
79.	hydroxybutanyl-trcaffeoylhexaric acid ^b^ 1	C_37_H_34_O_19_	781.1622	6.53	0.177	2	
80.	hydroxybutanyl-trcaffeoylhexaric acid ^b^ 2	C_37_H_34_O_19_	781.1622	6.66	1.585	2	
81.	isobutanyl-tricaffeoylhexaric acid ^b^	C_37_H_34_O_18_	765.1672	8.54	1.755	2	
82.	2-methyllbutanyl/ isovaleryl-tricaffeoylhexaric acid ^b^	C_38_H_36_O_18_	779.1829	9.21	1.531	2	
**Flavonoids**							
83.	myricetin *O*-hexoside	C_21_H_20_O_13_	479.0831	4.55	0.159	2	[9,26]
84.	myricetin *O*-acetylhexoside	C_23_H_22_O_14_	521.0939	5.19	0.483	2	[9]
85.	hyperoside ^a^	C_21_H_20_O_12_	463.0885	5.29	0.520	1	[9,26]
86.	luteolin-7-*O*-glucoside ^a,b^	C_21_H_20_O_11_	447.0934	5.31	0.213	1	
87.	patuletin *O*-hexoside ^b^	C_22_H_22_O_13_	493.0987	5.50	0.783	2	
88.	quercetin *O*-acetylhexoside ^b^ 1	C_23_H_22_O_13_	505.0988	5.61	0.666	2	
89.	nepetin *O*-hexoside ^b^	C_22_H_22_O_12_	447.1038	5.68	−0.061	2	
90.	kaempferol 3-*O*-glucoside ^a^	C_21_H_20_O_11_	447.0934	5.88	0.258	1	[9,26]
91.	isorhamnetin 3-*O*-glucoside ^a^	C_22_H_22_O_12_	477.1038	6.03	−0.124	1	[9,26]
92.	quercetin *O*-acetylhexoside isomer ^b^ 2	C_23_H_22_O_13_	505.0988	6.06	−0.859	2	
93.	quercetin *O*-malonylhexoside	C_24_H_22_O_15_	549.0886	6.06	−4.722	2	[9]
94.	6-methoxykaempferol *O*-hexoside ^b^	C_22_H_22_O_12_	477.1038	6.25	−1.340	2	
95.	kaempferol *O*-acetylhexoside	C_23_H_22_O_12_	489.1038	6.28	0.308	2	[9]
96.	hispidulin *O*-hexoside ^b^	C_22_H_22_O_11_	461.1089	6.32	0.792	2	
97.	jaceosidine *O*-hexoside ^b^	C_23_H_24_O_12_	491.1195	6.50	1.875	2	
98.	quercetin *O*-coumaroylhexoside isomer	C_30_H_26_O_14_	609.1250	7.07	0.643	2	[9,26]
99.	quercetin *O*-feruloylhexoside ^b^	C_31_H_28_O_15_	639.1355	7.20	0.308	2	
100.	quercetin *O*-coumaroylhexoside 2	C_30_H_26_O_14_	609.1250	7.30	0.643	2	[9,26]
101.	luteolin ^a,b^	C_15_H_9_O_7_	285.0406	7.58	−0.952	1	
102.	quercetin ^a^	C_15_H_10_O_7_	301.0354	7.62	−0.252	1	[26]
103.	kaempferol *O*-coumaroylhexoside 1	C_30_H_26_O_13_	593.1301	7.69	1.039	2	[9,26]
104.	patuletin ^b^	C_16_H_12_O_8_	331.0464	7.72	−0.304	2	
105.	herbacetin methyl ether	C_16_H_12_O_7_	315.0510	7.76	0.679	2	[9]
106.	nepetin ^b^	C_16_H_12_O_7_	315.0510	7.76	0.679	2	
107.	isorhamnetin *O*-*p*-coumaroylhexoside isomer ^b^ 1	C_31_H_28_O_14_	623.1406	7.88	0.981	2	
108.	kaempferol *O*-coumaroylhexoside isomer 2	C_30_H_26_O_13_	593.1301	7.95	1.072	2	[9,26]
109.	kaempferol *O*-caffeoylhexoside ^b^	C_30_H_26_O_14_	609.1250	7.99	0.364	2	
110.	isorhamnetin *O*-*p*-coumaroylhexoside isomer ^b^ 2	C_31_H_28_O_14_	623.1406	8.08	1.174	2	
111.	isorhamnetin *O*-caffeoylhexoside ^b^	C_31_H_28_O_15_	639.1355	8.12	0.496	2	
112.	axillarin ^b^ (quercetagetin 3,6-dimethylether)	C_17_H_14_O_8_	345.0616	8.19	0.027	2	
113.	hispidulin ^b^	C_16_H_12_O_6_	299.0561	8.84	0.430	2	
114.	kaempferol ^a^	C_15_H_9_O_7_	285.0406	8.84	−0.531	1	[26]
115.	naringenin	C_15_H_12_O_5_	271.0612	8.86	0.468	1	[9]
116.	isorhamnetin ^a^	C_16_H_12_O_7_	315.0510	9.12	0.013	1	[9,26]
117.	jaceosidin ^b^ (6-hydroxyluteolin-6,3’-dimethyl ether) ^a,b^	C_17_H_14_O_7_	329.0677	9.40	0.286	1	
118.	gnaphaliin 1	C_17_H_14_O_6_	313.0718	9.52	0.443	2	[9,26]
119.	quercetagetin-3,6,3’(4’)-trimethyl ether ^b^ 1	C_18_H_16_O_8_	359.0772	9.66	0.444	2	
120.	quercetin 7,3’(4’)-dimethyl ether ^b^	C_17_H_13_O_7_	329.0667	9.71	0.073	2	
121.	kaempferid/isokampferid	C_16_H_12_O_6_	299.0561	9.99	−0.561	2	[9]
122.	quercetagetin-3,6,3’(4’)-trimethyl ether ^b^ 2	C_18_H_16_O_8_	359.0772	11.18	−0.002	2	
123.	pinocembrin	C_15_H_12_O_4_	255.0663	11.63	−1.302	2	[9,26]
124.	galangin	C_15_H_10_O_5_	269.0457	11.74	−0.954	2	[26]
125.	gnaphaliin 2	C_17_H_14_O_6_	313.0718	12.19	0.123	2	[9,26]
126.	galangin methyl ether	C_16_H_12_O_5_	283.0612	12.27	−0.342	2	[9]
**Pyrones** (**phloroglucinol alpha-pyrones**)							
**№**	**Subclass** (**tentatively identified compound**)	**Molecular formula**	**Exact mass** **[M-H]** ^−^	**t_R_** (**min**)	**Δ ppm**	**Level of confidence**	**References**
127.	ethylpyrones A ^b,c^	C_22_H_26_O_8_	417.1555	13.15	0.429	3	
128.	ethylpyrones ^b,c^	C_22_H_26_O_9_	433.1504	13.46	−0.475	3	
129.	ethylpyrones ^b,c^	C_24_H_28_O_8_	443.1711	14.15	−0.341	3	
130.	isopropylpyrones ^b,c^	C_23_H_28_O_9_	447.1661	14.52	−0.147	3	
131.	methylpyrones (arenol)	C_21_H_24_O_7_	387.1449	14.65	0.087	2	[25]
132.	ethylpyrones ^b,c^	C_27_H_34_O_8_	485.2181	14.89	0.472	3	
133.	unknown ^b,c^	C_20_H_26_O_6_	361.1657	15.07	0.937	3	
134.	ethylpyrones ^b,c^	C_23_H_28_O_9_	447.1661	15.23	−0.370	3	
135.	ethylpyrones (arzanol)	C_22_H_26_O_7_	401.1606	15.76	−0.140	2	[25]
136.	ethylpyrones ^b,c^	C_27_H_34_O_8_	485.2181	15.81	0.101	3	
137.	ethylpyrones ^b,c^	C_24_H_30_O_9_	461.1817	16.30	0.012	3	
138.	ethylpyrones B ^b,c^	C_25_H_32_O_8_	459.2024	17.01	−0.307	3	
139.	isopropylpyrones ^b,C^	C_23_H_28_O_7_	415.1751	17.05	0.033	3	
140.	ethylpyrones ^b,c^	C_22_H_26_O_8_	417.1555	17.11	−0.146	3	
141.	ethylpyrones ^b,c^	C_23_H_28_O_7_	415.1751	17.81	0.177	3	
142.	ethylpyrone ^b,c^	C_24_H_28_O_8_	443.1711.	17.89	−0.476	2	
143.	1-methyl-propylpyrones ^b,c^	C_24_H_30_O_7_	429.1919	18.55	−0.458	3	
144.	isopropylpyrone ^b,c^	C_23_H_28_O_8_	431.1711	19.06	−0.489	3	
145.	unknown ^b,c^	C_23_H_32_O_6_	403.2126	19.22	0.128	3	
146.	ethylpyrones ^b^ (6-O-desmethyl-auricepyrone)	C_24_H_30_O_7_	429.1919	19.57	−0.458	2	[24,27]
147.	ethylpyrones ^b,c^	C_23_H_28_O_8_	431.1711	19.86	−0.976	3	
148.	methylpyrones ^b^	C_26_H_32_O_7_	455.2075	20.19	0.513	3	[8,9,30]
149.	1-methyl-propylpyrones (heliarzanol/isobar)	C_24_H_30_O_8_	445.1868	20.39	−0.744	3	
150.	1-methyl-propylpyrones ^b,c^	C_24_H_28_O_7_	427.1762	20.42	−5.914	3	
151.	ethylpyrones ^b^ (23-methyl-6-O-desmethylauricepyrone)	C_25_H_32_O_7_	443.2075.	20.73	−0.060	3	[24,27]
152.	ethylpyrones (heliarzanol/isobar)	C_24_H_30_O_8_	445.1868	21.00	−0.182	2	[8,9]
153.	ethylpyrones ^b^	C_27_H_34_O_7_	469.2232	21.23	0.050	2	[30]
154.	ethylpyrones B ^b,c^	C_25_H_32_O_8_	459.2024	21.67	0.368	3	
155.	ethylpyrones ^b,c^	C_27_H_34_O_8_	485.2181	21.97	−0.023	3	
156.	ethylpyrones C ^b,c^	C_24_H_28_O_7_	427.1762	22.13	0.102	2	
157.	isopropylpyrones ^b^	C_28_H_36_O_7_	483.2388	22.41	−0.055	3	[24]
158.	ethylpyrones C ^b,c^	C_24_H_28_O_7_	427.1762	23.01	−0.038	2	
**Other compounds**							
159.	gnaphaliol *O*-hexoside	C_19_H_24_O_9_	395.1348	8.14	−6.316	2	[9]
160.	micropyrone	C_14_H_20_O_4_	251.1289	9.12	−1.045	*2*	[8]
161.	Italipyrone 1	C_22_H_24_O_7_	399.1449	17.33	−1.369	2	[9]
162.	Italipyrone 2	C_22_H_24_O_7_	399.1449	20.39	−2.231	2	[9]
**№**	**Identified/tentatively annotated compound**	**Molecular formula**	**Exact mass** **[M+H]^+^**	**t_R_** (**min**)	**Δ ppm**	**Level of confidence**	**References**
163.	6-hydroxytremeton	C_13_H_14_O_3_	219.1013	7.21	−1.373	3	[31]
164.	2-isobutyryl-6-acetylprenylphloroglucinol	C_17_H_22_O_6_	323.1481	13.16	−2.522	3	[32]
165.	2-isobutyryl-4-prenylphloroglucinol	C_15_H_20_O_4_	265.1431	17.68	−1.379	3	[32]
166.	2-metylvaleryl-4-prenylphloroglucinol	C_17_H_24_O_4_	293.1741	18.28	−2.066	3	[32]

^a^ Identified by comparison with an authentic standard; ^b^ reported for the first time in *H. italicum*; ^c^-undescribed in the literature; A, B, and C compounds labeled with the same capital letters share the same fragmentation patterns; level of confidence: 1—compound identified by comparison to the reference standard; 2—putatively annotated compound; and 3—putatively characterized compound classes.

[M-H]^−^ at *m/z* 417.156 (**127** and **140**), 431.171 (**147**), 445.187 (**152**), and 459.203 (**138** and **154**) indicated an additional oxygen atom than **135**, **141**, **146**, and **151**, respectively. The characteristic transitions in both negative and positive ion modes related to the losses of *m/z* 58.043 Da (C_3_H_6_O) and 72.058 Da (C_4_H_8_O) were employed to depict hydroxyprenyl residue. As an example, **147** with [M-H]^−^ at *m/z* 431.171 (consistent with C_23_H_27_O_8_) yielded fragment ions at *m/z* 247.097 [M-H-C_9_H_10_O_3_-H_2_O]^−^, 207.065 [M-H-C_9_H_10_O_3_-C_3_H_6_O]^−^, and 193.050 [M-H-C_9_H_10_O_3_-C_4_H_8_O]^−^ originating from the hydroxyprenyl moiety in the phloroglucinol skeleton (Appendix A). In (+) ESI-MS, the key points for the assignment of hydroxyprenyl moiety were the fragment ions at *m/z* 361.129 [M+H-C_4_H_8_O]^+^, 189.055 [M+H-C_8_H_10_O_3_-C_4_H_8_O]^+^, and 177.054 [M+H-C_9_H_10_O_3_-C_4_H_8_O]^+^ together with the abundant ions at *m/z* 261.112 [M+H-C_8_H_10_O_3_-H_2_O]^+^ (100%) and 249.112 [M+H-C_9_H_10_O_3_-H_2_O]^+^ (90%). Additionally, prominent fragment ions at *m/z* 153.054 and 109.064 in (-) ESI/MS and 167.070 and 155.070 in (+) ESI/MS indicated an ethyl group at C-11 of the pyrone skeleton (Figure 4). The aforementioned structures are consistent with hydroxyprenyl residue at C-3, as was found in Heliarzanol [8,24]. It is worth noting that the 4-hydroxypyrone core exists in two tautomeric forms. On the other hand, the rotamers arise from the intramolecular hydrogen bonds between the oxygen functions of pyrone moiety and phenolic hydroxyl groups vicinal to the methylene bridge. Thus, different rotameric and/or tautomeric forms occur [8]. In the same way, hydroxyprenyl residue was evidenced in **138** and **154** ([M-H]^−^ at *m/z* 459.203), where 2-methyl-oxobutyl residue at C-1 was deduced (Appendix A).

MS data of **128** showed [M-H]^−^ at 433.150 (consistent with C_22_H_25_O_9_) (Appendix A). In (-) ESI-MS, the precursor ion gave a base peak at *m/z* 267.087 [M-H-C_9_H_10_O_3_]^−^, indicating the same α-pyrone moiety as in arzanol (**135**) and a phloroglucinol unit with two oxygen atoms more than arzanol (Figure 4). The subsequent losses of 88.053 Da (C_3_H_6_O_2_) and 74.038 Da (C_4_H_8_O_2_) indicated the presence of a dihydroxylated prenyl unit. Accordingly, **128** yielded abundant fragment ions at *m/z* 223.0969 [M-H-C_9_H_10_O_3_-CO_2_]^−^, 193.050 [M-H-C_9_H_10_O_3_-C_3_H_6_O_2_]^−^, and 179.034 [M-H-C_9_H_10_O_3_-C_4_H_8_O_2_]^−^, suggesting a hydroxyl group at both terminal C-19 and C-20 in the prenyl residue (Appendix A). This assumption was confirmed by the fragmentation pattern in (+) ESI-MS, where indicative ions at *m/z* 347.112 [M+H-C_4_H_8_O_2_]^+^, 193.0496 [M+H-C_8_H_10_O_3_-C_4_H_8_O_2_]^+^, and 181.044 [M-H-C_9_H_10_O_3_-C_4_H_8_O_2_]^−^ were generated (Appendix A). By analogy, compound **134** with [M-H]^−^ at *m/z* 447.166 (calc. for C_23_H_28_O_9_, −0.370 ppm) possessed an additional methylene group in the phloroglucinol unit in comparison with **128** and oxopropyl moiety at C-1. Compounds **128** and **134** were not reported in the literature.

MS/MS spectra of **153** with [M-H]^−^ at *m/z* 469.223 was acquired (calc. for C_27_H_33_O_7_)—there was a mass difference of 68 Da (C_5_H_8_) between its phloroglucinol moiety and that of arzanol (Appendix A). The prominent fragment ions at *m/z* 191.034 [M-H-C_9_H_10_O_3_-C_8_H_16_]^−^, 179.034 [M-H-C_9_H_10_O_3_-C_9_H_16_]^−^, and 166.026 [M-H-C_9_H_10_O_3_-C_10_H_17_]^−^ indicated a geranyl chain (C_10_H_17_) at C-3 instead of a prenyl one in arzanol (Appendix A). This structure was consistent with the compound previously identified in *H. decumbens* [30]. The subsequent losses of 112.126 Da (C_8_H_16_), 124.126 Da (C_9_H_16_), and 137.134 Da (C_10_H_17_) are indicative of a geranyl unit. In the same way, the two compounds **136** and **155** with [M-H]^−^ at *m/z* 485.218 (calc. for C_27_H_33_O_8_) gave an abundant fragment ion at *m/z* 319.155 (72.4%) and the subsequent transitions 319.155→191.034, 319.155→179.034 and 319.155→166.026 indicating the losses of 128.121 Da (C_8_H_16_O), 140.121 Da (C_9_H_16_O), and 153.129 Da (C_10_H_17_O) (Appendix A). Accordingly, hydroxygeranyl residue was suggested at C-3 of both pyrones.

An isopropyl group at C-11 was dedicated to compounds **130**, **139**, **144**, and **157** (Appendix A). Concerning **139** ([M-H]^−^ at *m/z* 415.176, calc. for C_23_H_27_O_7_), the MS/MS spectrum afforded prominent fragment ions at *m/z* 247.097 [M-H-C_9_H_12_O_3_]^−^ and *m/z* 235.097 [M-H-C_10_H_12_O_3_]^−^, indicating an addition methylene group in the pyrone residue in comparison with **135** (arzanol). The presence of a prenyl moiety at C-3 was also evidenced by the abundant fragment ions at *m/z* 361.127 [M+H-C_4_H_8_]^+^, 193.0496 [M+H-C_10_H_12_O_3_-C_4_H_8_]^+^, and 181.049 [M+H-C_9_H_12_O_3_-C_4_H_8_]^+^ in (+) ESI-MS. Thus, the isopropyl group was suggested at C-11, as was observed in Helitalone B [33] (Figure 4). The aforementioned compound is an isobar of compound Arenol B previously isolated from *H. oocephalum* [24]. The MS/MS spectra of **144** with [M-H]^−^ at *m/z* 431.177 (calc. for C_23_H_28_O_8_) and **130** with [M-H]^−^ at *m/z* 447.166 (calc. for C_23_H_28_O_9_) suggested one additional oxygen atom in **144** and two supplementary oxygen atoms in **130** in comparison with **139**, respectively. The hydroxyprenyl moiety in **144** was dedicated from the indicative transitions in (+) ESI-MS 433.185→265.107→193.049 and 433.185→253.106→163.039 (Appendix A).

By analogy with **153**, geranyl moiety at C-3 of **157** (M-H]^−^ at *m/z* 483.23, calc. for C_28_H_36_O_7_) was evidenced, as was observed in achyroclinopyrone C, previously isolated from *H. oocephalum* [24] (Appendix A).

Two compounds, **131** with [M-H]^−^ at *m/z* 387.145 (calc. for C_21_H_24_O_7_) and **148** with [M-H]^−^ at *m/z* 455.208 (calc. for C_26_H_32_O_7_), shared the same methylpyron moiety evidenced by the base peak [M-H-C_8_H_8_O_3_]^−^ at *m/z* 235.097 and 303.160, respectively (Appendix A). The only difference was attributed to the occurrence of geranyl moiety (C_10_H_17_) in **148** instead of a prenyl moiety (C_5_H_9_) in **131**. In (+) ESI-MS/MS, **148** afforded abundant fragment ions at 333.090 [M+H-C_9_H_16_]^+^ (52.3%) and 181.050 [M+H-(α-pyrone-C)-C_9_H_16_]^−^ (100%), suggesting a geranyl moiety at C-3. Thus, **131** was consistent with arenal, previously identified in *H. italicum* [34], while **148** was ascribed as phloroglucinol pyron isolated from *H. decumbens* [30] (Figure 4).

The extracted ion chromatograms of prenylated phloroglucinol α-pyrones demonstrated that the *H. italicum* aerial parts’ profile was dominated by arzanol (**135**) (19.28%), ethylpyrone **153** (11.34%), 6-*O*-desmethyl-auricepyrone (**146**) (10.72%), together with ethylpyrone **151** (8.14%), and isopropylpyrones (**139**) (5.30%) (Appendix A).

### 2.7. Other Compounds

The known monomer pyrone micropyrone **160** with [M-H]^−^ at *m/z* 251.129 (calc. for C_14_H_19_O_4_) was dereplicated together with italipirone and its isomer (**161**/**162**) consisting of both pyrone and a benzofurane ring [6]. A pyrone moiety was evidenced by the transition 399.144→233.0814 resulting from the loss of ethylpyrone as was observed in the heterodimer-pyrones (Appendix A).

Four compounds were tentatively annotated in positive ion mode. Compound **163**, [M+H]^+^ at *m/z* 219.101 (C_13_H_14_O_3_), gave fragment ions at *m/z* 209.091 [M+H-H_2_O]^+^, 176.096 [M+H-H_2_O-CO]^+^, and 159.080 [M+H-H_2_O-CO-CH_2_]^+^ and a base peak at m/z 183.080 [M+H-2H_2_O]^+^. Further fragmentation pathways of **163** consisted of the structure of 6-hydroxytremetone, previously isolated from *H. umbraculigerum* [31]. MS/MS spectrum of **165** (C_15_H_20_O_4_) revealed consecutive losses of three OH groups at *m/z* 247.132 [M+H-H_2_O]^+^, *m/z* 191.070 [M+H-C_4_H_8_-H_2_O]^+^, and *m/z* 173.059 [M+H-C_4_H_8_-2H_2_O]^+^, consisting of the presence of a phloroglucinol skeleton. In addition, a base peak at m/z 209.080 [M+H-C_4_H_8_]^+^, together with fragment ions at *m/z* 163.075 [M+H-C_4_H_8_-H_2_O-CO]^+^ and 135.080 [M+H-C_4_H_8_-H_2_O-2CO]^+^, were attributed to the structure of 2-isobutyryl-4-prenylphloroglucinol, previously found in *H. asperum*, *H. flanaganii*, *H. gymnocoum,* and *H. infaustum* [32]. Similar fragmentation pathways with losses of three OH groups, C_4_H_8_, and CO groups were presented in the MS/MS spectrum of **166** (C_17_H_24_O_4_). Moreover, **164** differed from **165** with an additional methylene group (C_2_H_4_) and was tentatively annotated as 2-metylvaleryl-4-prenylphloroglucinol, isolated from *H. caespititium* [32]. In the same way, **164** gave a base peak at *m/z* 249.075 [M+H-H_2_O-CO-C_4_H_8_]^+^ and the subsequent fragment ions, corresponding to the losses of H_2_O and CO groups. [M-H]^−^ at *m/z* 323.148 (calc. for C_17_H_22_O_6_) indicated 58 Da higher than that of **135**, suggesting additional acetyl moiety (Appendix A). Compounds **164**–**166** belong to the groups of acylphloroglucinols. The predominant secondary metabolites among other compounds were micropyrone (**160**) (63.34%), followed by 6-hydroxytremeton (**163**) (14.15%) and 2-isobutyryl-6-acetylprenylphloroglucinol (**164**) (10.96%).

UHPLC-HRMS analysis revealed that phloroglucinol alpha-pyrones was the major group of secondary metabolites in *H. italicum* extract and reach up to 38.05% of all of the 166 compounds. Acylquinic acids (29.52%) and flavonoids (13.57%) were also found in high quantities in the studied species. Among all compounds, arzanol (**135**) (7.33%) was the predominant metabolite in the extract, while the amount of 3,4-dicaffeoylquinic acid (**51**), 3,5-dicaffeoylquinic acid (**52**), and 1,5-dicaffeoylquinic acid were found to be 5.61%, 5.58%, and 4.58%, respectively. The structures of the main annotated metabolites in *H. italicum* extract are presented in Figure 5.

### 2.8. Antioxidant and Cholinesterase Inhibitory Activity

In the present study, *H. italicum* extract was tested for antioxidant and cholinesterase inhibitory potential (Table 2). DPPH and ABTS+ were used to evaluate radical scavenging ability, while the reduction abilities were calculated by the CUPRAC, FRAP, and phosphomolybdenum (PHMD) methods. The metal chelating method was based on the binding of transition metals by phytochemicals. Results are presented as trolox equivalents and ethylenediaminetetraacetic acid (EDTA), and *H. italicum* extract revealed high activity of all of the used antioxidant methods. The enzyme inhibitory properties of *H. italicum* extracts were examined against both AChE and BChE. The results are calculated as a Galantamine (Gal) equivalent. The studied extract showed average AChE (1.64 ± 0.09 mg GALAE/g) and low BChE inhibitory potential (0.11 ± 0.02 mg GALAE/g). A high selectivity of the enzyme inhibitory activity of *H. italicum* extract targeting AChE was demonstrated.

## 3. Passive Avoidance Test

In the passive avoidance test, the ability of the animals to learn the new task was assessed on the fifth day (Figure 6). On this day, only the group treated with the combination galantamine and *H. italicum* extract showed a slight increase in the latency time in comparison to the control and other groups. On day 12, when the memory of the animals was evaluated, groups treated with *Ginkgo biloba* and with the combination galantamine and *H. italicum* showed a statistically significant increase (*p* ≤ 0.001) in the latency times compared to the control group (Figure 7). The result indicated an improvement of the memory processes after 12 days of administration of these compounds.

The incidence of dementia, a disease strongly associated with cognitive impairment, is on the rise globally with expectations that the number of patients will double every 20 years [35]. Early prevention of dementia is critical because no definitive therapy has been established. One of the promising sources for the prevention and treatment of different types of dementia, incl. Alzheimer’s disease (AD), are plant sources [36]. In addition to evoking an antioxidant response, *H. italicum* essential oil also displays neuroprotective effects on mental fatigue/burnout, which generates further interest in *Helichrysum* extracts as potential cognitive-enhancing agents. Based on previous investigations, the *H. italicum* methanol–aqueous extract is worth investigating for the memory-ameliorating effects. We hypothesized that the combination of the *H. italicum* extract with the classical AChE inhibitor galantamine would support an improvement of the learning and memory process in the behavioral test in mice. To consider this hypothesis, we used a passive avoidance test to investigate whether 12 days of per oral administration of the combination lead to an improvement of the memory processes in comparison to the control group, single drug, or plant extract application. Such a combination could be used in the future for the development of specific cognitive-enhancing formulations.

When the mice were treated with the combination of galantamine (3 mg/kg) and *H. italicum* extract (200 mg/kg), a significant difference between combined and single applications was observed. It is worth noting that the combination was more beneficial for the memory-enhancement process in comparison with *G. biloba* (EGb 761) extract (*p* ≤ 0.001) (Figure 6).

Taking into consideration that the antioxidant response is a key point in the memory-ameliorating effects, we proved the antioxidant activity of the *H. italicum* extract by chemical-based assays based on the scavenging activity toward a stable free radical (DPPH and ABTS), the reduction of metal ions (FRAP and CUPRAC), metal chelating, and total antioxidant potential [37]. The main mechanism by which antioxidants play their protective role included hydrogen atom transfer (HAT) or a single electron transfer (SET). Often, more complex reactions like mixed HAT/SET, stepwise electron transfer-proton transfer, concerted electron–proton transfer, or sequential proton loss electron transfer occur [38]. The studied *H. italicum* extract demonstrated strong radical scavenging, metal-reducing, and chelating activity. Taking into account that the DPPH scavenging activity depends on hydrogen atom transfer (HAT); the phenolic compounds (hydroxycinnamic and acylquinic acids, and flavonoids) are very active due to the lower bond dissociation energies (BDE) of the phenolic hydroxyl groups [39]. Consistent with already published studies, we confirm a positive correlation between these phenolic compounds’ classes in *H. italicum* and the antioxidant potential [15]. On the other hand, *H. italicum* extract demonstrated a moderate inhibitory activity towards acetylcholinesterase.

In line with the aforementioned results, we suggest that the main phenolic compounds belonging to hydroxycinnamic caffeoylquinic and acylhexaric acids, phloroglucinol derivatives, and some flavonoids (luteolin, quercetin, pinocembrin, naringenin) in *H. italicum* may hold significance for the memory enhancement observed in the passive avoidance test in mice.

Previously, caffeic, coumaric, and sinapic acids have been reported to improve cognitive function [40,41,42]. In different studies, the listed metabolites have been shown to suppress the breakdown of the amyloid-precursor protein, which is pathologically related to the Aβ (1–42) protein, lipid peroxidation, and neurite extension of hippocampal neurons [41,42]. These findings suggest that hydroxycinnamic acid intake may account for the improvement of the cognitive function, but the exact mechanism by which this intake affects cognitive function in humans is currently unknown. In the study of Kato et al. [43] on a small group of community-dwelling elderly individuals with complaints of subjective memory loss, after a 6-month intake period of caffeoylquinic acid there was significant improvement in attentional, executive, and memory functions.

Studies using mice and cultured neurons have shown that chlorogenic acid protect neurons and suppress the aggregation of amyloid beta (Aβ—one of the main hallmarks of AD) through antioxidant effects [41,44]. In addition, there are reports on beneficial effects of chlorogenic acid and/or its derivatives for ameliorating of spatial learning and memory and reducing behavioral deficits in a variety of in vivo animal models of disease or behavior [45].

Our previous study on *Achillea* species (Asteraceae family) has shown that the antioxidant activity of the species could be related to the synergetic action of metabolites such as 4-hydroxybenzoic acid-hexoside, 3-feruloylquinic acid, 1,3-dicaffeoylquinic acid, caffeic acid *O*-hexoside, 5-p-coumaroylquinic acid, and isorhamnetin 3-*O*-glucoside [12]. In this context, the bioinformatics analysis conducted through the platform Swiss Target Prediction confirmed that phenolic compounds, axillarin, quercetagetin-3,6,3′(4′)-trimethyl ether, quinic acid, dicaffeoylquinic acids, chlorogenic acid, and hesperetin could interact with target proteins, including hydrolase, electrochemical transporters, and transcription factors, involved in neuromodulation and neuroprotection [18].

Concerning flavones and flavonols, luteolin have been proved to exert a suppressive effect against endoplasmic reticulum stress activation and inflammatory signaling pathways in AD animal and cell models. Thus, the compound alleviates the learning and memory impairment in mice [46]. Numerous studies investigated the neuroprotective effects of quercetin in the central nervous system, especially in multiple in vitro and in vivo models of AD. Nakagawa et al. [47] reviewed seven studies of animal experiments estimating the neuroprotective effect of quercetin, in which quercetin improved cognition and memory deficits in rodent animal models of AD. The possible protective mechanisms of quercetin mainly involved the inhibition on Aβ aggregation and tauopathy, the anti-oxidative and anti-inflammatory activity, and amelioration of mitochondrial dysfunction [48].

A previous investigation demonstrated that the flavanone pinocembrin can be used to treat diseases such as stroke, AD, and vascular dementia [49,50]. Kang et al. revealed that pinocembrin attenuated learning and memory deficits induced by vascular dementia, by inducing the expression of Reelin, apoER2, and p-dab1 in the hippocampus. These authors also demonstrated that pinocembrin improved the impaired learning ability in rats by reducing the number of errors and decreasing the latency to step down in the step-down type of passive avoidance test [51]. Moreover, another flavanone naringenin dose dependently improved spatial recognition memory in Y maze, the discrimination ratio in a novel object discrimination task, and retention and recall capabilities in a passive avoidance test in the lipopolysaccharide-induced cognitive decline in rats. The authors suggested that naringenin have improved retention and recall in passive avoidance test via affecting synaptic plasticity [52]. Naringenin could ameliorate learning and memory deficit in passive avoidance tests in neurotoxic conditions through improvement of hippocampal oxidative stress and neuronal injury and increase the expression level of choline acetyltransferase [53].

It is worth noting that prenylated phloroglucinol α-pyrones and acylphloroglucinols could also attribute to the memory enhancing potential of *H. italicum* extract. It has been found that phloroglucinol reduced oxidative stress induced by oligomeric Aβ1–42 (Aβ1–42) in the HT-22 hippocampal cell line. In addition, the reduction in dendritic spine density caused by either hydrogen peroxide or Aβ1–42 was significantly rescued by phloroglucinol in rat primary hippocampal neuron cultures. Furthermore, phloroglucinol attenuated memory deficits in the 5XFAD mouse model of AD based on the Morris water maze and T-maze tests. As a whole, phloroglucinol displays a therapeutic potential for AD patients as a ROS-scavenger [54].

Herein, the most prominent effect of memory was found in the group treated with *H. italicum* extract combined with galantamine. In our previous study, galantamine successfully reverses scopolamine-induced memory impairment in mice, especially on the 12th day [55]. A characteristic that makes galantamine appropriate for the treatment of AD is a selective inhibitory activity on the enzyme acetylcholinesterase (AChE) in the central nervous system with a small effect on peripheral tissues [56]. Subsequently to galantamine’s approval for the treatment of mild-to-moderate AD in 2001, a wide variety of species have been assessed in pursuit of new AChE inhibitors [57]. Gonçalves et al. [58] reported that the methanol extract of *H. italicum*, rich with phenolic compounds (caffeoylquinic and dicaffeoylquinic acids, and pinocembrin), showed high inhibitory activity against enzymes involved in Alzheimer’s disease like AChE, tyrosinase, and α-glucosidase.

Our results confirm the well-known positive effect of *G. biloba* extract on the memory processes. After 12 days of administration, the group treated with *G. biloba* (EGb 761) statistically significantly prolonged the latency time in the passive avoidance test in comparison to the control group. A standardized extract of *G. biloba* leaves EGb 761 is a popular dietary supplement taken to enhance mental focus and used for treatment of certain cerebral dysfunctions and dementias associated with aging and AD [59].

The extract EGb 761 is known to contain about 24% flavonoids and 6% terpene lactones. There is reliable evidence that standardized Ginkgo extract exhibits several molecular and cellular neuroprotective mechanisms, including attenuation of apoptosis, inhibition of membrane lipid peroxidation, anti-inflammatory effects, and direct inhibition of Aβ aggregation. There is also data showing that *G. biloba* extract significantly inhibits the activity of AChE in the brain [60]. The positive effects of EGb 761 on memory function, including in stress situations, have been demonstrated in many experiments involving mice, rats, and even chicks while using various tests and paradigms, including conventional passive and active avoidance, Morris water maze, scopolamine-induced amnesia, learned helplessness, and olfactory learning ability, etc. These studies involve acute or chronic treatment with EGb and highlight the positive effect concerns on short- or long-term memory [61].

## 4. Materials and Methods

### 4.1. Chemicals

Galantamine hydrobromide (Sopharma, Sofia, Bulgaria, Mw = 368.3 g/mol, purity > 98%); *Ginkgo biloba* capsules 80 mg (Adipharm, Sofia, Bulgaria, dry standardized leaf extract (EGb 761), contains 6% terpenolactones and 24% flavonoid glycosides).

Acetonitrile (hypergrade for LC-MS), formic acid (for LC-MS), and methanol (analytical grade) were purchased from Chromasolv (Sofia, Bulgaria). The reference standards used for compound identification were obtained from Extrasynthese (Genay, France) for protocatechuic, gentisic, *p*-coumaric, *m*-coumaric, o-coumaric, syringic, vanillic, and salicylic acids and hyperoside, luteolin 7-*O*-glucoside, isorhamnetin 3-*O*-glucoside, luteolin, quercetin, kaempferol, isorhamnetin, and jaceosidin. Neochlorogenic, chlorogenic, caffeic, 3,4-dicaffeoylquinic, 3,5-dicaffeoylquinic, and 1,5-dicaffeoylquinic acids were supplied from Phytolab (Vestenbergsgreuth, Germany).

### 4.2. Plant Material

*H. italicum* ssp. *italicum* aerial parts were collected in the full flowering stage in July 2022 at the Botanical Garden of Medicinal Plants (BGMP), Wroclaw Medical University, Poland, and kindly supplied by Prof. Dr. Adam Matkowski. A voucher specimen was deposited at Herbarium Academiae Scientiarum Bulgariae (SOM 178 491). Subsequently, the plant material was dried at room temperature.

### 4.3. Sample Extraction

Air-dried powdered aerial parts (50 g) were extracted with 80% MeOH (1:20 *w*/*v*) by sonication (100 kHz, ultra-sound bath Biobase UC-20C) for 15 min (×2) at room temperature. Then, the methanol was evaporated in vacuo and water residues were lyophilized (lyophilizer Biobase BK-FD10P) to yield crude extract 1.38 g. Afterwards, the lyophilized extract was dissolved in 80% methanol (0.1 mg/mL), filtered through a 0.45 μm syringe filter (Polypure II, Alltech, Lokeren, Belgium), and an aliquot (2 mL) of each solution was subjected to UHPLC–HRMS analyses. The same extract was used for further in vitro and in vivo tests.

### 4.4. UHPLC-HRMS Profiling

The phytochemical analyses were performed on a Q Exactive Plus mass spectrometer (ThermoFisher Scientific, Inc. Walthham, USA). The apparatus operated in negative and positive modes in *m/z* range from 100 to 1000. The chromatographic separation was performed on a reversed phase column Kromasil EternityXT C18 (1.8 µm, 2.1 × 100 mm) at 40 °C. The chromatographic analyses were performed as previously described [14]. Separation was achieved on an UHPLC system Dionex Ultimate 3000RSLC (ThermoFisher Scientific, Inc.) The mobile phase consisted of A: water (with 0.1% formic acid) and B: acetonitrile (with 0.1% formic acid). The used gradient was as follows: 5% B for 1 min, gradually turned to 30% B over 19 min, increased gradually to 50% B over 5 min, increased gradually to 70% B over 5 min, and finally increased gradually to 95% over 3 min. The system was then turned to the initial condition and equilibrated over 4 min. The flow rate was 300 μL/min and the injection volume was 1 μL. Data acquisition and processing were performed with Xcalibur 4.2 software (ThermoScientific, Walthham, USA). Peaks annotations were based on accurate masses in full MS and ddMS2, MS/MS fragmentation pathways, precursor and fragment ions relative abundance, elemental composition, comparison with the retention times, fragment spectra, and chromatographic behavior of reference standards obtained from an in-house database of previously identified compounds. The compounds identified with reference standards during the present study belong to confidence class 1, while the compounds that were putatively annotated belong to level 2 (reported in *H. italicum* previously), and putatively characterized classes belong to level 3 [29].

MZmine 2 software was applied to the UHPLC–HRMS raw files of the studied *H. italicum* extracts for the semi-quantitative analysis. Results are expressed as the % peak area of the compound to the total peak areas of the corresponding group secondary metabolites and all metabolites.

### 4.5. Antioxidant and Enzyme Inhibitory Assays

DPPH radical scavenging assay: The dsample solution (1 mg/mL; 1 mL) was added to 4 mL of a 0.004% methanol solution of DPPH. The sample absorbance was measured at 517 nm after a 30 min incubation [62].

ABTS radical scavenging assay: Briefly, ABTS+ was produced by reacting 7 mM ABTS solution with 2.45 mM potassium persulfate and allowing the mixture to stand for 12–16 h in the dark at room temperature. The ABTS solution was diluted with methanol to an absorbance of 0.700 ± 0.02 at 734 nm. The sample solution (1 mg/mL; 1 mL) was added to the ABTS solution (2 mL) and mixed. The sample absorbance was measured at 734 nm after a 30 min incubation at room temperature [62].

CUPRAC assay: The sample solution (1 mg/mL; 0.5 mL) was added to the reaction mixture containing CuCl**_2_** (1 mL, 10 mM), neocuproine (1 mL, 7.5 mM), and NH**_4_**Ac buffer (1 mL, 1 M, pH 7.0). Similarly, a blank was prepared without CuCl**_2_**. Then, the sample and blank absorbances were measured at 450 nm after a 30 min incubation at room temperature. The absorbance of the blank was subtracted from that of the sample [62].

FRAP (ferric reducing antioxidant power) activity assay: The sample solution (1 mg/mL; 0.1 mL) was added to premixed FRAP reagent (2 mL) containing acetate buffer, 2,4,6-tris(2-pyridyl)-S-triazine (TPTZ) (10 mM) in 40 mM HCl and ferric chloride (20 mM) in a ratio of 10:1:1 (*v*/*v*/*v*). Then, the sample absorbance was read at 593 nm after a 30 min incubation at room temperature. DPPH, ABTS, CUPRAC, and FRAP activity were expressed as milligrams of trolox equivalents (mg TE/g extract) [62].

Metal chelating activity assay: Briefly, the sample solution (1 mg/mL; 2 mL) was added to FeCl2 solution (0.05 mL, 2 mM). The reaction was initiated by the addition of 5 mM ferrozine (0.2 mL). Similarly, a blank was prepared without ferrozine. Then, the sample and blank absorbances were measured at 562 nm after 10 min incubation at room temperature. The metal chelating activity was expressed as milligrams of EDTA (disodium edetate) equivalents (mg EDTAE/g extract) [62].

Phosphomolybdenum method: The sample solution (1 mg/mL; 0.3 mL) was mixed with 3 mL of reagent solution (0.6 M sulfuric acid, 28 mM sodium phosphate and 4 mM ammonium molybdate). The sample absorbance was measured at 695 nm after a 90 min incubation at 95 °C. The total antioxidant capacity was expressed as millimoles of trolox equivalents (mmol TE/g extract) [62].

Cholinesterase inhibitory activity assay: The sample solution (1 mg/mL; 50 μL) was mixed with DTNB (5,5-dithio-bis(2-nitrobenzoic) acid, Sigma, St. Louis, MO, USA), (125 μL) and AChE (acetylcholines-terase (Electric ell AChE, Type-VI-S, EC 3.1.1.7, Sigma)), or BChE (BChE (horse serum BChE, EC 3.1.1.8, Sigma, Burlington, MA, USA)) solution (25 μL) in Tris–HCl buffer (pH 8.0) in a 96-well microplate and incubated for 15 min at 25 °C. The reaction was then initiated with the addition of acetylthiocholine iodide (ATCI, Sigma) or butyrylthiocholine chloride (BTCl, Sigma) (25 μL). Similarly, a blank was prepared by adding the sample solution to all reaction reagents without an enzyme (AChE or BChE) solution. The sample and blank absorbances were read at 405 nm after 10 min incubation at 25 °C. The absorbance of the blank was subtracted from that of the sample, and the cholinesterase inhibitory activity was expressed as galanthamine equivalents (mgGALAE/g extract).

### 4.6. Animals

The experiment was conducted on 30 male mice, line H, with a body weight in the range of 28–32 g. All experiments were approved by the Institutional Animal Care Committee at the Medical University of Sofia. The animals were divided into 5 groups (n = 6). The tested substances were administered perorally (p.o.) in the following doses:Group 1: control group, receiving distilled water only;Group 2: animals treated with *H. italicum* extract—200 mg/kg;Group 3: animals treated with galantamine—3 mg/kg;Group 4: animals treated with *Ginkgo biloba* (EGB761)—100 mg/kg;Group 5: animals treated with galantamine (3 mg/kg) and *H. italicum* extract—200 mg/kg.

The substances were in a solid form; they were ground, and the required amount of distilled water was added.

All treatments were performed for 12 days. During this period, animals were observed daily for behavioral changes and signs of toxicity. The experiment was conducted in accordance with the Directive 2010/63/EU of the European Parliament and of the Council on the protection of animals used for scientific purposes (No. 346 of 28 February 2023) from the Bulgarian Food Safety Agency.

### 4.7. Passive Avoidance Test

The learning and memory processes of the experimental mice were evaluated by a passive avoidance test using an automated shuttle box (Gemini Avoidance System, San Diego Instruments). The apparatus consists of two identical compartments (25 × 20 × 16 cm each) with an electrified grid floor. The chambers are separated by a wall with a guillotine door (8 × 6 cm). The test started with an acclimatization period of 20 s. The maximum duration of the trial was 300 s. If a mouse entered the dark chamber, a weak electric shock with an intensity of 0.5 mA and a duration of 3 s. was applied through the grid floor. Learning was assessed by the latency of entry into the dark compartment. An increase in latency time indicated an improvement of the learning and memory processes. The tested compounds were administered p.o. 1 h before passive avoidance testing on days 1–5 of the experiment and then daily for 7 days without testing in the apparatus. On the 12th day the memory storage, processes were evaluated again by passive avoidance test.

### 4.8. Statistical Analysis

The results from the passive avoidance test were processed statistically using GraphPad Prism 6 software and presented as mean ± SEM. The groups were compared using one-way ANOVA followed by a post-hoc comparison of sample means (Tukey test). The differences were considered statistically significant at *p* ≤ 0.05.

## 5. Conclusions

For the first time, more than 90 secondary metabolites were reported in *H. italicum* including phenylethanoid glycosides, a series of hydroxybenzoic and hydroxicinnamic acid–glycosides, caffeoyl-hydroxydihydrocaffeoylquinic, *p*-coumaroyl-caffeoylquinic and tricaffeoylquinic acids, malonyl-dicaffeoylquinic acids, a series of caffeoylhexaric acids, and methoxylated derivatives of scutelarein, quercetagetin, and 6-hydroxyluteolin. The main finding of the phytochemical study was a comprehensive profiling of heterodimer-pyrones where 23 compounds, undescribed in the literature, were annotated. This study is the first attempt to propose the fragmentation patterns of four subclasses pyrones in LC-HRMS. For the first time, the cognitive-enhancing properties of *H. italicum* extract is reported. We have demonstrated that the combination of the extract and classical acetylcholinesterase inhibitor galantamine significantly improved the learning and memory after 12 days administration in a passive avoidance test in mice. The effect is more pronounced even than that of *G. biloba* (EGb 761). Taken together, our data suggest significant antioxidant activity of *H. italicum* extract by radical scavenging activity, reducing power and metal chelating capacity. The extract showed moderate acetylcholinesterase and low butyrylcholinesterase inhibitory potential. It appears that powerful antioxidant activity coupled with moderate and selective AChE inhibitory ability accounted for the cognitive-enhancing potential of *H. italicum* extract. Prenylated phloroglucinol derivatives, acylquinic and acylhexaric acids, and flavonoids may hold significance for the memory improvement potential of *H. italicum* extract. Further analysis concerning mechanisms of action is needed to advance our knowledge on the pharmacological effects of *H. italicum*.

## Figures and Tables

**Figure 1 plants-12-02755-f001:**
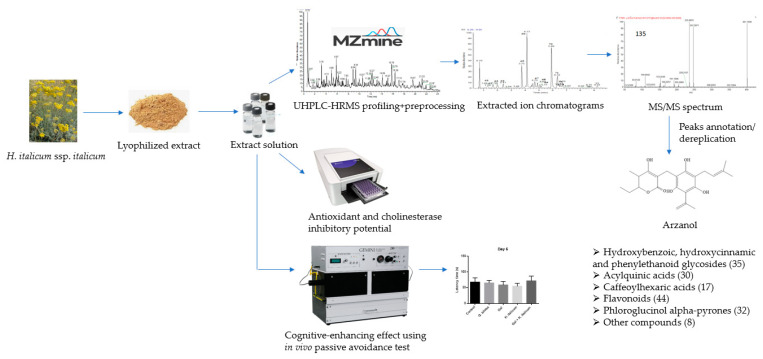
A flow chart for the design of the experiment.

**Figure 2 plants-12-02755-f002:**
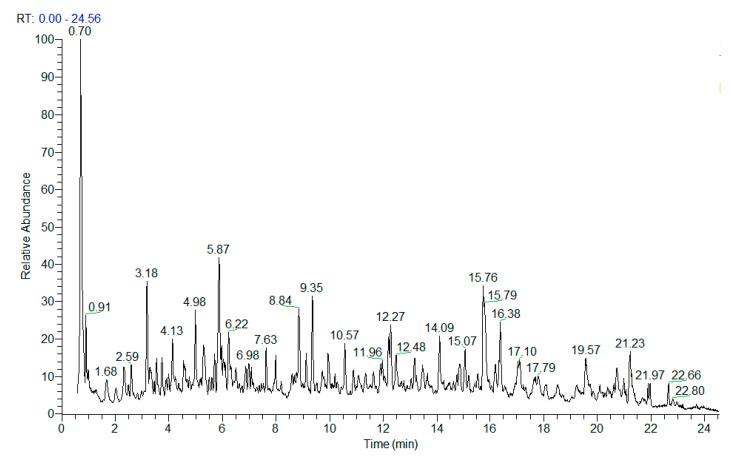
TIC in negative ion mode of *H. italicum* extract.

**Figure 3 plants-12-02755-f003:**
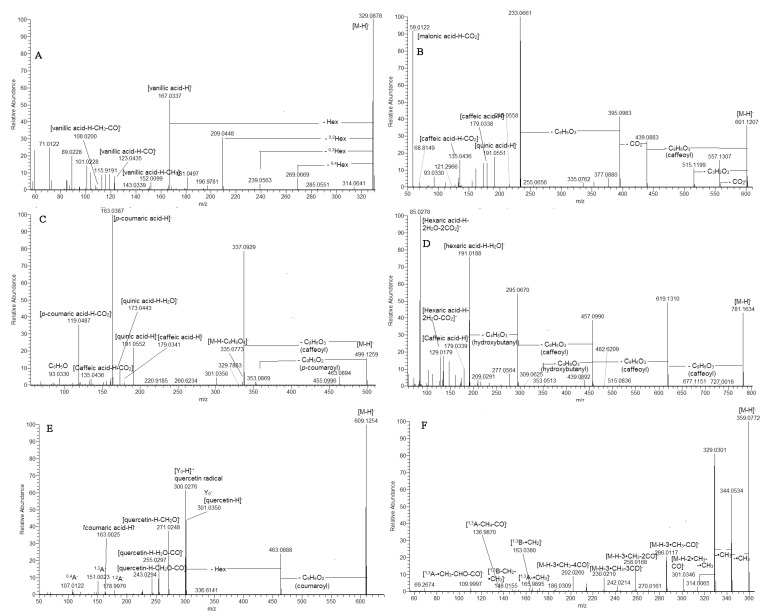
MS/MS spectra of vanillyl O-hexose (**11**) (**A**), 1,3-dicaffeoylquinic acid malonyl (**53**) (**B**), 3-*p*-coumaroyl-5-caffeoylquinic acid (**59**) (**C**), hydroxybutanyl-trcaffeoylhexaric acid (**80**) (**D**), quercetin *O*-coumaroylhexoside (**100**) (**E**), and quercetagetin-3,6,3’(4’)-trimethyl ether (**119**) (**F**).

**Figure 4 plants-12-02755-f004:**
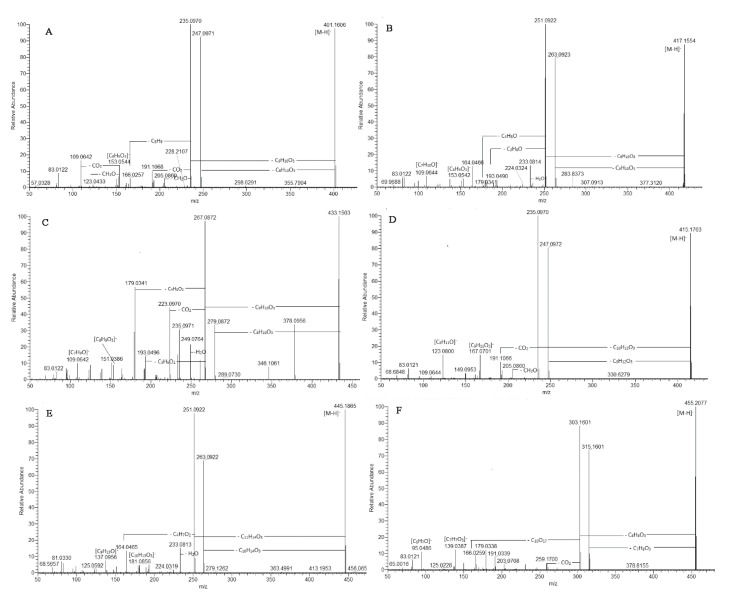
MS/MS spectra of phloroglucinol alpha-pyrones; compounds **135** (**A**), **140** (**B**), **128** (**C**), **139** (**D**), **149** (**E**), and **148** (**F**).

**Figure 5 plants-12-02755-f005:**
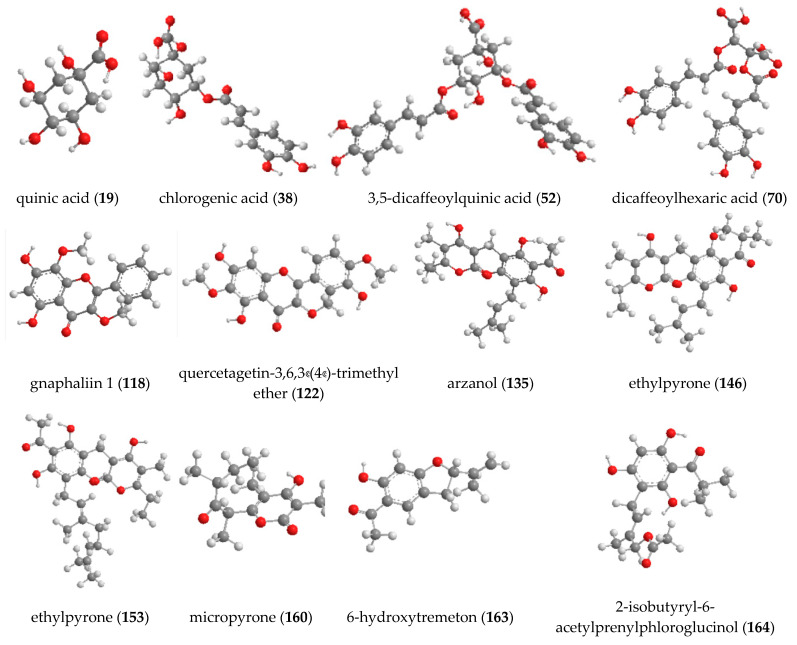
Chemical structure of the main compounds in *H. italicum* extract.

**Figure 6 plants-12-02755-f006:**
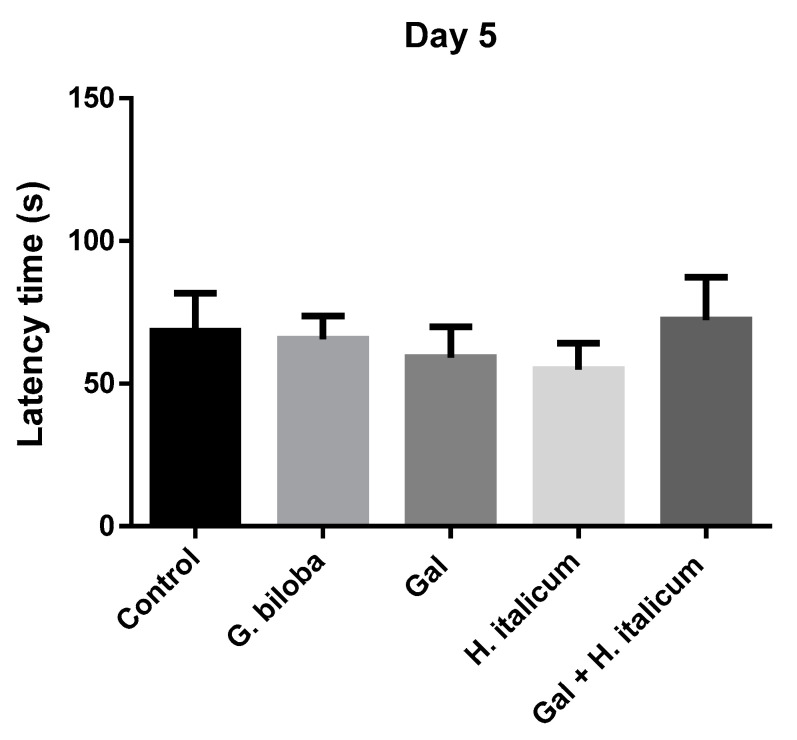
Effect of the compounds on the latency time on the 5th day of the administration. Results are presented as mean ± SEM. There are no statistically significant differences.

**Figure 7 plants-12-02755-f007:**
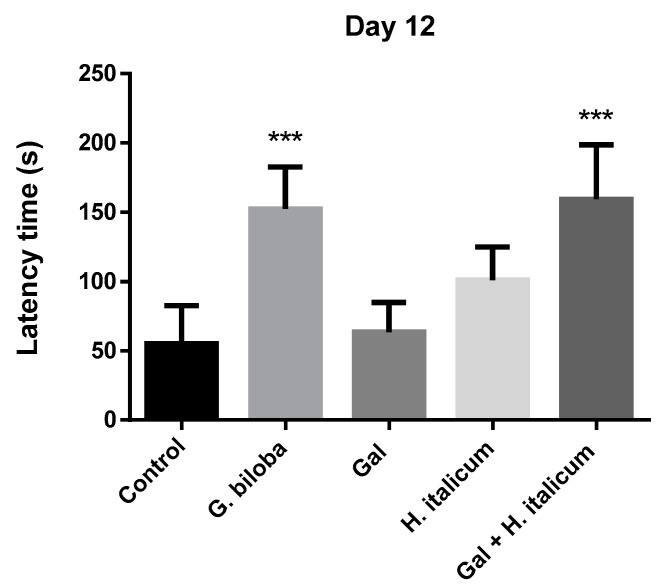
Effect of the compounds on the latency time on the 12th day of the administration. Results are presented as mean ± SEM. *** *p* ≤ 0.001 statistically significant increase in the latency time in groups treated with *Ginkgo biloba* and Gal+ *H. italicum* extract in comparison to the control group.

**Table 2 plants-12-02755-t002:** Antioxidant activity of *H. italicum* extract.

Activity	Means ± SD
DPPH (mg TE/g)	110.33 ± 3.47
ABTS (mg TE/g)	234.70 ± 5.21
CUPRAC (mg TE/g)	354.23 ± 17.51
FRAP (mg TE/g)	210.24 ± 8.68
Chelating (mg EDTAE/g)	44.32 ± 0.75
Phosphomolybdenum (mmol TE/g)	1.97 ± 0.06

## Data Availability

Not applicable.

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
