# Peer review of "Phytochemical Profiling, Antioxidant and Cognitive-Enhancing Effect of *Helichrysum italicum* ssp. *italicum* (Roth) G. Don (Asteraceae)"

_plants, 2023, doi:10.3390/plants12152755_

Round 1

Reviewer 1 Report

This manuscript (plants-2517813) found that Helichrysum italicum ssp. italicum extract exhibited antioxidant activity, improved memory and learning processes, and revealed over 90 newly identified secondary metabolites. These findings highlight the plant as a potential source of cognitive-enhancing substances.

The sections in general are well written, needing only to check the statistics present in the figures, as well as making the hypothesis and the objectives of the work clearer. The materials and methods section can be improved by adding specific information about the measurement conditions, step by step. Perhaps a flowchart could improve the idea about the experiment? Add statistics as well as the sample size. However, overall, it is a good job of chemical characterization.

There is a large amount of data in the form of tables, which could be included in supplementary material. In addition, I suggest to the authors, to perform a modelling in order to demonstrate the chemical characterization in 3D of the molecular geometry. Why were only these analysis and characterization methods chosen?

What were the statistics used for comparison, in figures 2 and 3? Improve the description of the results as well as a more robust discussion, founded and demonstrating if the objectives as well as the hypothesis was accepted or refuted. In addition, make clearer, what is the hypothesis of the work?

L418 (p <0.001). check all manuscript common and dots;

Figure 2, 3. statistical?

The conclusion should be rewritten. It should clearly demonstrate the potential of your work. In addition, what are the future prospects?

Moderate editing of the English language needed. Check grammar, verbosity and spelling for more suitable and focused on the content in question. Additionally, consider revising the discussion section.

Author Response

Response to Reviewer 1 Comments

This manuscript (plants-2517813) found that Helichrysum italicum ssp. italicum extract exhibited antioxidant activity, improved memory and learning processes, and revealed over 90 newly identified secondary metabolites. These findings highlight the plant as a potential source of cognitive-enhancing substances.

The sections in general are well written, needing only to check the statistics present in the figures, as well as making the hypothesis and the objectives of the work clearer. The materials and methods section can be improved by adding specific information about the measurement conditions, step by step. Perhaps a flowchart could improve the idea about the experiment? Add statistics as well as the sample size. However, overall, it is a good job of chemical characterization.

Response: Dear reviewer, thanks for the comment. The statistics present in the figures was checked (See Figures 5 and 6). The materials and methods section was changed according to the reviewer’s recommendation (See Materials and Methods). The hypothesis and the objectives of the work were presented clearer (See Introduction, row 105-111). According to the reviewer’s suggestion, a flowchart was embedded as Figure 1 in the manuscript.

There is a large amount of data in the form of tables, which could be included in supplementary material. In addition, I suggest to the authors, to perform a modelling in order to demonstrate the chemical characterization in 3D of the molecular geometry. Why were only these analysis and characterization methods chosen?

Response: Thanks for the comment. Concerning the compound annotation/dereplication, the table with precursor and fragment ions with relative abundances is provided in the Supplemental material. In the main document, the exact masses, m/z errors, retention times, level of confidence and available references are depicted in Table 1.

A new figure 5 was embedded into the manuscript with 3D formulas of the principal compounds in Helichrysum extract.

High resolution mass spectrometry (HRMS) coupled to ultra-high-performance liquid chromatography (UHPLC) and data dependent MS/MS analyses provide very valuable information on secondary metabolites for in-depth metabolite annotation studies (Allard et al., Curr opinion chem biol. 2017; 36:40-49). We elaborated metabolite profilings of several Asteraceae species based on rapid UHPLC-Orbitrap HRMS for metabolite annotation/dereplication (Gevrenova et al., Ind Crops Prod. 2020; 155: 112817, Ak et al., Food chem. toxicol, 2021; 153, 112268; Gevrenova et al., Antioxidants; 2021; 10: 1180; Gevrenova et al., Plants, 2022; 12: 22; Zheleva-Dimitrova et al., Plants 2023; 12: 1009). Recently, metabolic and biological profilings of T. macrophyllum, T. balsamita, T. vulgare, Telekia speciosa and Cicerbita alpina integrated with multivariate data analysis shed light into future potential industrial applications using the tested species.

Several different spectrophotometrical methods were used for the determination of the antioxidant activity of H. italicum extract. The chemical-based assays can be divided into different methods for assessing antioxidant activity: those based on the scavenging activity toward a stable free radical (DPPH and ABTS), the reduction of metal ions (FRAP and CUPRAC, Folin-Ciocalteu assay), metal chelating or total antioxidant potential (Rumpf et al., 2023). In this study, we used DPPH, ABTS, FRAP, CUPRAC, metal chelating and phosphomolybdenum methods. These assays were the most frequently used antioxidant assays for phenolic compounds analysis (See Results and discussion, row 487-497).

What were the statistics used for comparison, in figures 2 and 3? Improve the description of the results as well as a more robust discussion, founded and demonstrating if the objectives as well as the hypothesis was accepted or refuted. In addition, make clearer, what is the hypothesis of the work?

Response: Thanks for the comment. The statistics part was presented in the Material and Methods section and now is separated in individual part (See 3.8, row 741-745). The groups were compared using one-way ANOVA followed by a post-hoc test of Tukey. The differences were considered statistically significant at p≤ 0.05. The results and discussion were improved according to the reviewer’s recommendations (See Results and discussion).

L418 (p <0.001). check all manuscript common and dots;

Response: Thanks for the comment. The all manuscript was checked and corrected according to the reviewer’s recommendation.

Figure 2, 3. statistical?

Response: Thanks for the comment. Figures were changed and statistics were included (See Figures 5 and 6).

The conclusion should be rewritten. It should clearly demonstrate the potential of your work. In addition, what are the future prospects?

Response: Thanks for the comment. The conclusion was rewritten according to the reviewer’s recommendation, future prospects were included (See Conclusion, row 753-764).

Moderate editing of the English language needed. Check grammar, verbosity and spelling for more suitable and focused on the content in question. Additionally, consider revising the discussion section.

Response: Thanks for the comment. All manuscript was changed according reviewer’s recommendation.  The discussion was revised.

Reviewer 2 Report

1. In my opinion, Table S1 in supplementary materials is more suitable to be shown in manuscript because it demonstrates more information.

2. In present version, component identification is not readable enough. Some are simple and some are complicated. It is suggested that for each category of natural products, a typical component should be regarded as an example explaining and showing its fragmentation pathway by words and figures. If one is not typical enough, some others could be added. Fragmentation principles are more important. Some similar published papers could be more helpful.

3. In passive avoidance test, galantamine did not show considerable and significant effects. So, is it suitable as a positive drug?

4. Group 4: animals treated with Ginkgo biloba – 100 mg/kg. Ginkgo biloba extract (EGB761) is more suitable.

5. Galantamine and H. italicum extract were given to animals by what ways? Powder? Solution? How to prepare them?

Author Response

Response to Reviewer 2 Comments

  1. In my opinion, Table S1 in supplementary materials is more suitable to be shown in manuscript because it demonstrates more information.

Response: Dear review, thanks for the comment. Taking into account a large amount of data in the Table S1 on 16 pages, the exact masses, m/z errors, retention times, level of confidence and available references of the annotated compounds are depicted in Table 1 in the main document. In order to make compound annotation/dereplication more apprehensible and in line with the reviewer’s suggestion, Figure 3 was embedded illustrating (together with Fig. 4) the fragmentation patterns of representative compounds belonging to each group assigned. In addition, Figure 5 was created consisting of the 3D molecular formulas of the main secondary metabolites in the Helichrysum extract.

  1. In present version, component identification is not readable enough. Some are simple and some are complicated. It is suggested that for each category of natural products, a typical component should be regarded as an example explaining and showing its fragmentation pathway by words and figures. If one is not typical enough, some others could be added. Fragmentation principles are more important. Some similar published papers could be more helpful.

Response: Thanks for the comment. In response to the Reviewer’s suggestion, a new Figure 3 was embedded with representative fragmentation patterns as follows: vanillyl O-hexose (compound 11) for the sugar esters (hydroxybensoic acid derivatives), 1, 3-dicaffeoylquinic acid-malonyl (53) and 3-p-coumaroyl-5-caffeoylquinic acid (59) (acylquinic acids), hydroxybutanyl-tricaffeoylhexaric acid (80) (acylhexaric acids), quercetin-coumaroylhexoside (98) (flavonoid glycosides), quercetagetin-3,6, 4'-trimethyl ether (119) (methoxylated aglycones). Additionally, a figure with 3D formulas of the main secondary metabolites was also created. It’s worth noting that the profiling highlighted the presence of a large number of phloroglucinol α-pyrones. As this is the first attempt for in deep annotation of the pyrones by UHPLC-HRMS, a special focus of attention was made on this class metabolites (Fig. 4). Our group has elaborated metabolite profilings of several Asteraceae species based on rapid UHPLC-Orbitrap HRMS for metabolite annotation/dereplication where a systematic investigation on the fragmentation patterns and diagnostic fingerprints in the MS/MS spectra of acylquinic and acylhexaric acids, and flavonoids was developed (Gevrenova et al., Ind Crops Prod. 2020; 155: 112817, Ak et al., Food chem. toxicol, 2021; 153, 112268; Gevrenova et al., Antioxidants; 2021; 10: 1180; Gevrenova et al., Plants, 2022; 12: 22; Zheleva-Dimitrova et al., Plants 2023; 12: 1009). The aforementioned references were cited in the current manuscript, accordingly.

  1. In passive avoidance test, galantamine did not show considerable and significant effects. So, is it suitable as a positive drug?

Response: Thanks for the comment and question. We used galantamine as a positive control because it is from a plant origin, approved for the treatment of Alzheimer’s disease and there is a large information about its influence on the learning and memory in animals and humans. In the presented experiment, on day 5 there is no statistically significant difference among the groups. A possible reason for the results from day 12 may be that in experiments with living organisms, sometimes there may be a different result than expected and other groups may have a better result than galantamine in the particular case. We also cited our previous study, where galantamine successfully reverse the scopolamine-induced memory in mice, especially on the 12-th day.

  1. Group 4: animals treated with Ginkgo biloba– 100 mg/kg. “Ginkgo biloba extract (EGB761) is more suitable.

Response: Thanks for the comment. The chanced was done (See Materials and Methods, row 718).

  1. Galantamine and H. italicumextract were given to animals by what ways? Powder? Solution? How to prepare them?

Response: Thanks for the questions. The used substances were solid. They were ground and the required amount of distilled water was added. The tested substances were administered perorally (p.o.).

Reviewer 3 Report

See PDF

Author Response

Response to Reviewer 3 Comments

At the begining of introduction, you should give some general information about plant essential oils

Response: Dear reviewer, thanks for the comment.  The introduction was changed according to your recommendation (See Introduction, row 49-61).

I think a whole chromatogram of HPLC could be better

Response: Dear reviewer, thanks for the comment. A total ion chromatogram in negative ion mode was presented (See Figure 2).

What do you mean with (21) ?

Response: [21] is a reference for the Level of confidence (Sumner, L. W.; Amberg, A.; Barrett, D.; Beale, M. H.; Beger, R.; Daykin, C. A.; Fan, T. W.-M.; Fiehn, O.; Goodacre, R.; Griffin, J. L., Proposed minimum reporting standards for chemical analysis: chemical analysis working group (CAWG) metabolomics standards initiative (MSI). Metabolomics 2007, 3, 211-221.). According to CAWG, the annotation of compounds in LC-MS can be classified as follows: Level of confidence: 1-compound identified by comparison to reference standard; 2-putatively annotated compound; 3- putatively characterized compound classes (See notes under Table 1).

This table is less understandable, you added antioxidant methods with other cholinesterase inh. assays, please separate the different assays.

Response: Dear reviewer, thanks for the comment. Table 2 was changed according your recommendation.

The font in the table is different from the text _

Response: Dear reviewer, thanks for the comment. The font of the Figure 6 was changed.

Reviewer 4 Report

The manuscript entitled “ Phytochemical Profiling, Antioxidant and Cognitive-enhancing effect of Helichrysum italicum ssp. italicum (Roth) G. Don 3 (Asteraceae)describes a study that uses a highly sensitive technique to identify a large number of compounds from different classes of compounds. The manuscript is very well laid out, the confirmation of the structures is supported by experimental data richly. The antioxidant activity of the extract has been proven, which is logical considering the compounds contained in it. More interesting is the fact that the combined application of the extract and galangamine achieves the effect of Ginkgo biloba.

I have only some technical remarks to the authors.

It would be better if AChE and BChE  were written with the full name in the abstract.

Line 60 - hydroxybeonzoic acids should be hydroxybenzoic acids.

Lines 171, 327, 345 Only Table is written in brackets. Please specify which table you mean - table 1 or table 1S.

Author Response

Response to Reviewer 4 Comments

The manuscript entitled “ Phytochemical Profiling, Antioxidant and Cognitive-enhancing effect of Helichrysum italicum ssp. italicum (Roth) G. Don 3 (Asteraceae)” describes a study that uses a highly sensitive technique to identify a large number of compounds from different classes of compounds. The manuscript is very well laid out, the confirmation of the structures is supported by experimental data richly. The antioxidant activity of the extract has been proven, which is logical considering the compounds contained in it. More interesting is the fact that the combined application of the extract and galangamine achieves the effect of Ginkgo biloba.

I have only some technical remarks to the authors.

It would be better if AChE and BChE  were written with the full name in the abstract.

Response: Dear reviewer, thanks for the comment. The full names are written in the Abstract (See Abstract, row 20).

Line 60 - hydroxybeonzoic acids should be hydroxybenzoic acids.

Response: Thanks for the comment. The correction was done.

Lines 171, 327, 345 Only Table is written in brackets. Please specify which table you mean - table 1 or table 1S.

Response: Thanks for the comment. All corrections were done.

Round 2

Reviewer 1 Report

I thank the authors for responding to my comments and questions. I believe that the manuscript (plants-2517813) can be accepted for publication.

Minor changes.